# Proteomic characterization of intrahepatic cholangiocarcinoma identifies risk-stratifying subgroups and EIF4A1 as a therapeutic target

Tilman Werner [1,2,3,9], Johanna Thiery[1,2,4,9], Klara-Luisa Budau[1], Annika Topitsch [1,2,3], Miguel Cosenza-Contreras[1,2], Niko Pinter [1], Frank Hause [1,8], Julius Rühlmann[1], Gaia Gentile[5], Jannis Heyer[1], Konrad Kurowski [1], Julia Schüler [5], Philipp Anton Holzner[6], Martin Werner[1,4], Carlie Sigel[7], Laura H. Tang[7], Peter Bronsert [1,10] & Oliver Schilling [1,4,10] ✉

Intrahepatic cholangiocarcinoma (ICC) features poor survival due to frequent recurrences and limited prognostic markers. Using mass spectrometry-based proteomics, we analyze two independent cohorts comprising 80 and 62 treatment-naive ICC tumors, along with 9 independent patient-derived xenografts (PDX). In the first cohort, we identify two subclusters with distinct times-to-recurrence (TTR): An extracellular matrix (ECM)-enriched cluster (mean TTR 859 days) and a proliferation cluster (mean TTR 229 days). A 4-protein classifier trained on our cohort accurately stratifies these clusters in the Dong et al. dataset (2022) and in our second cohort, revealing similar proteomic motifs and clinical outcomes. The translation regulator EIF4A1, enriched in ICCs of both clusters, emerges as a therapeutic target, as its inhibition with eFT226 significantly reduces tumor growth in an ICC PDX model. Proteomic analyses of various PDX models also emphasize the critical role of tumor-stroma interactions in ICC. Overall, this study establishes two prognostic proteomic clusters, validates their relevance across datasets, and highlights EIF4A1 inhibition as a potential therapeutic strategy.

Cholangiocarcinomas (CCA) are the second most prevalent primary hepatic tumors and comprise 15% of all liver malignancies. Depending on their location along the bile ducts, CCAs are categorized as intrahepatic (ICC), perihilar or distal adenocarcinomas. CCAs possess different genetic, histologic, and clinical features, but all are characterized by a late onset with diffuse clinical symptoms, frequent recurrences and dismal overall survival[1–3].

[1]Institute for Surgical Pathology, Medical Center—University of Freiburg, Faculty of Medicine—University of Freiburg, Freiburg, Germany. [2]Faculty of Biology, University of Freiburg, Freiburg, Germany. [3]Spemann Graduate School of Biology and Medicine (SGBM), University of Freiburg, Freiburg, Germany. [4]German Cancer Consortium (DKTK), Partner Site Freiburg, a Partnership Between German Cancer Research Center (DKFZ) and University Medical Center Freiburg, Freiburg, Germany. [5]Charles River Laboratories Germany GmbH, Freiburg, Germany. [6]Department of General and Visceral Surgery, Medical Center—University of Freiburg, Faculty of Medicine—University of Freiburg, Freiburg, Germany. [7]Department of Pathology, Memorial Sloan Kettering Cancer Center, New York, NY, USA. [8]Present address: Department of Pharmaceutical Chemistry and Bioanalytics, Institute of Pharmacy and Institute of Molecular Medicine, Martin Luther University Halle-Wittenberg, Halle (Saale), Germany. [9]These authors contributed equally: Tilman Werner, Johanna Thiery. [10]These authors jointly supervised this work: Peter Bronsert, Oliver Schilling. ✉e-mail: oliver.schilling@uniklinik-freiburg.de

ICCs are rare cancers, nevertheless their incidence rate is rising worldwide[3]. Risk factors include bile duct stasis, inflammatory diseases of the liver, obesity and diabetes, parasitic or viral infections, and alcohol or tobacco abuse[4,5]. Remarkably, approximately 50% of cases occur without identifiable risk factors[6].

ICCs arise in small bile ducts and ductules inside the liver. Diagnoses are frequently delayed due to nonspecific symptoms, imprecise imaging criteria, and a lack of reliable, non-invasive tumor markers[1,2,7]. Consequently, at the moment of primary diagnosis, patients often show an advanced tumor stage and are not eligible for surgery[2,7]. For a remaining minority, an extensive surgical resection represents the first-line treatment option[5,8]. However, up to 70% of resected patients develop recurrences and standardized treatment guidelines for adjuvant chemo-, targeted-, or immunologic therapies are rare or entirely absent[5-7,9-13]. In addition, the benefit of adjuvant chemo- or radiotherapy is nowadays considered minimal[8,10,12-15]. To enable successful tumor re-resections and personalized therapies, robust prognostic markers are essential.

Pathological assessment of resected ICC adheres to the criteria of the American Joint Committee on Cancer (AJCC) and the Union for International Cancer Control (UICC)[2,16,17]. Previously, we identified lymph node- and lymphangio-invasion, as well as AJCC tumor staging,

as potential prognostic markers for the ICC cohort presented here[18]. Important histopathological markers for ICC include cytokeratins (KRTs) −7, −17, −19 and MUC1, while KRT20 and napsin A are typically missing[19-21]. More precise markers for tumor subtyping and predicting ICC recurrence are lacking (Fig. 1a).

Due to its rarity, ICC remains insufficiently described on a molecular level. Of note, comprehensive omics-based studies are scarce[2,3]. Nevertheless, genomic and transcriptomic studies have described the importance of *FGFR* fusions and *IDH1, ARID1A, BAP1, KRAS*, and *TP53* mutations for ICC tumorigenesis. Additionally, several publications proposed different molecular classifications into fibrotic, immune-deserted, or innate- and acquired immunity-mediated inflammatory subtypes on a genomic and transcriptomic level[22-27]. The first integrated proteogenomic profile elaborating on ICC subtypes was presented by Dong et al.[28], and was followed by subsequent studies on cohorts collected at the same hospital[29,30].

In recent years, interest in the proteomic characterization of tumors has increased, spurred by groundbreaking studies from the Clinical Proteomic Tumor Analysis Consortium (CPTAC)[31]. Since transcriptome alterations are only partially reflected at the protein level and cannot represent processes such as proteolytic events or other post-translational modifications, proteomic profiling has become an

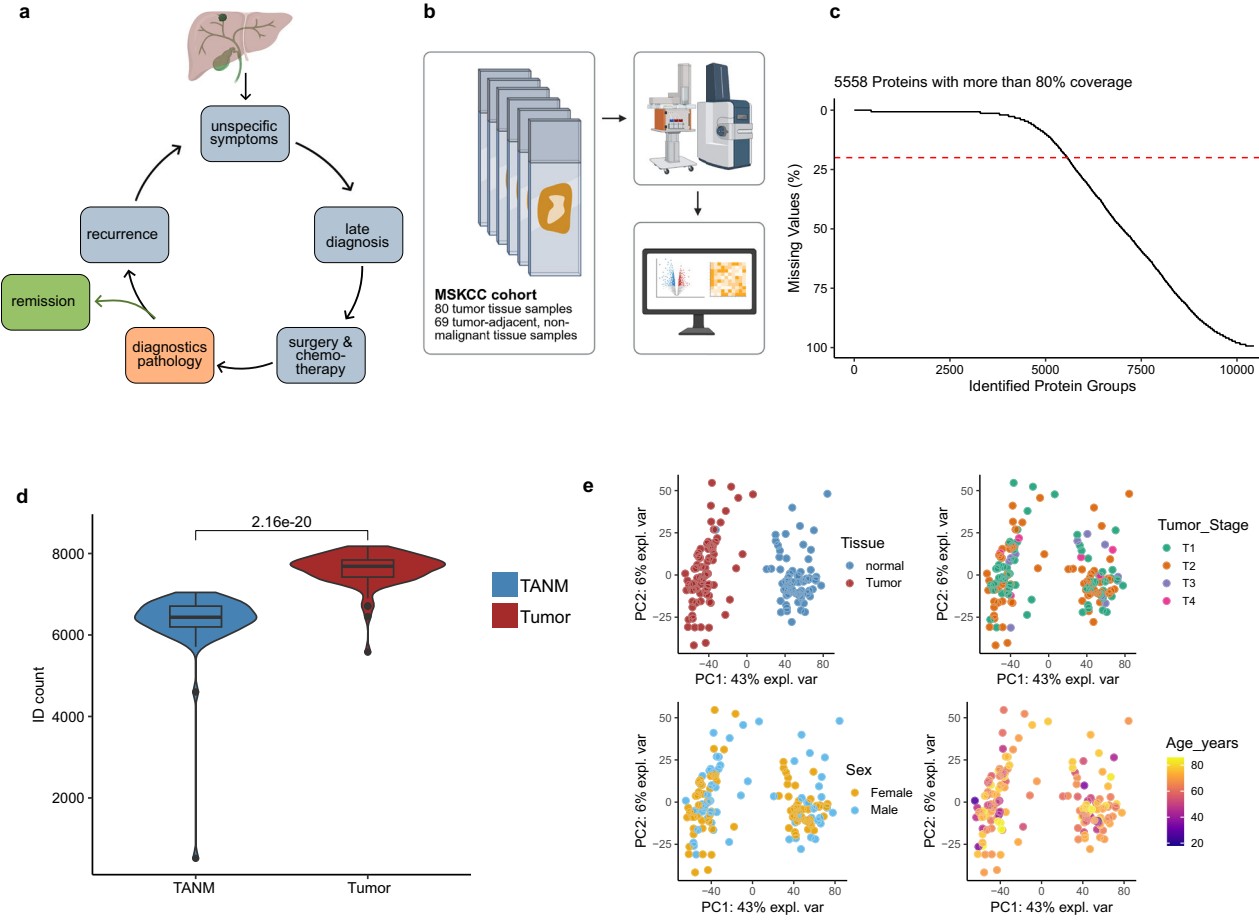

**Fig. 1 | Cohort overview. a** Unspecific symptoms, late diagnoses and a lack of prognostic markers complicate treatment of ICC. Partially created in BioRender. Thiery, J. (2026) https://BioRender.com/dv8sa68. **b** Tumor and tumor-adjacent, non-malignant (TANM) tissue from 80 patients were macrodissected, measured, and analyzed. Created in BioRender. Schilling, O. (2026) https://BioRender.com/7ehes0l. **c** Percentage of missing values across all detected proteins. The red dotted line indicates 20% missing values. **d** Protein-ID numbers across all samples (*n* = 149) of the cohort were compared between TANM and tumor tissue (unpaired *t*-test,

*p* < 0.0001). Boxplots show median (center line), interquartile range (IQR, extending from the 1st to the 3rd quartile, box), and 1.5 IQR (whiskers). Tumor tissue in red, TANM tissue in blue. **e** Principal Component Analysis (PCA) of the entire cohort colored by tissue type (tumor in red, TANM in blue), tumor stage (T1 in green, T2 in orange, T3 in purple, T4 in pink), sex (female in orange, male in blue), and age (lowest age in purple, highest age in yellow). Source data are provided as a Source Data file.

integral component in understanding cancer biology[32]. Advances in methods such as data-independent acquisition (DIA) have facilitated cohort-wide studies with deep proteome coverage[33].

Here, we present the proteomic characterization of ICC samples from the Memorial Sloan Kettering Cancer Center (MSKCC), comprising 80 treatment-naive ICC tumors and 69 patient-matched, tumor-adjacent non-malignant tissues (TANM), highlighting tumor-specific proteome alterations. We identified two predominant proteomic subtypes of ICC, which are strongly associated with recurrence-free survival. Using a machine learning-based classification model, we transferred these clusters to an independent cohort from the University Medical Center Freiburg (UKF), consisting of 62 treatment-naive tumors with 61 matching TANM tissues, and to the dataset published by Dong et al.[28], uncovering similar proteomic effects. Further analyses of proteolytic activity and patient-derived xenograft models (PDX) emphasize the importance of tumor-stroma interactions in ICC progression. Finally, we introduce EIF4A1 as a potential drug target and assess the effect of its novel inhibitor eFT226 (Zotatifin) on ICC growth in PDX.

## Results and discussion
### Description of the cohorts
Our MSKCC-ICC cohort comprised 80 treatment-naive tumors and 69 patient-matched, tumor-adjacent non-malignant tissues (TANM) from a total of 80 ICC patients. Patient characteristics are provided in Supplementary Table 1 and Supplementary Data 1. Our independent UKF-ICC cohort included 62 treatment-naive tumors and 61 patient-matched TANM tissues; patient characteristics are summarized in Supplementary Table 2. The UKF-ICC cohort is solely used for validation of the subtype-specific classifier. The size of our ICC cohorts is smaller than the Fudan University ICC (FU-ICC) cohort, for which comprehensive proteogenomic characterizations have recently been published by Dong et al. (262 patients), Lin et al. (255 patients), and Deng et al. (217 patients)[28–30]. However, disease etiology and incidence are assumed to differ across world regions, and, to our knowledge, this study represents one of the first large proteomic analyses of ICC in Western countries[2,3,34].

### Proteome coverage
Across the MSKCC-ICC cohort, on average, 7020 proteins were identified per sample through DIA-PASEF measurements on a Bruker TimsTOF Flex mass spectrometer. Tumor tissues showed significantly higher identification numbers, perhaps due to more complex proteomic profiles (unpaired $t$-test, $p < 0.0001$, Fig. 1d Supplementary Data 2). We have used DIA-NN 1.9.2 for data analysis. More than 5500 proteins presented with less than 20% missing values throughout the entire cohort and were used for statistical analysis (Fig. 1c). Six outliers were reevaluated by a pathologist, reclassified as hepatocellular carcinomas or fibrosis, and removed from the study, yielding the aforementioned numbers of 80 ICC tumors and 69 TANM tissues.

### Tumor and TANM tissues show highly divergent proteome profiles
Principal component analysis and hierarchical clustering of the MSKCC cohort demonstrate highly divergent proteomic profiles of ICC and TANM proteomes (Fig. 1e and Supplementary Fig. 1A). To identify differentially regulated proteins in ICC vs. TANM, we employed linear models, which revealed 4865 proteins that were differentially regulated in the MSKCC-ICC cohort ($p < 0.01$, with FDR correction) (Fig. 2a and Supplementary Data 3).

Reassuringly, cytokeratins 7 (KRT7) and 19 (KRT19), commonly used markers to differentiate ICC from hepatocellular carcinoma, as well as epithelial marker melanoma cell adhesion molecule (MCAM/CD146), were strongly upregulated in tumors (KRT7 logFC = 4.03, $p_{adj} = 2.23E^{-36}$; KRT19 logFC = 4.75, $p_{adj} = 5.01E^{-56}$; MCAM logFC = 1.85,

$p_{adj} = 7.99E^{-37})^{19-21,35}$. Likewise, biglycan (BGN), an extracellular matrix proteoglycan frequently upregulated in tumor tissue and tumor microenvironment, was detected with higher abundance in ICC tissues (logFC = 2.02, $p_{adj} = 1.86E^{-28})^{36-38}$.

We sought to identify common biological themes within the ICC-associated proteins. To this end, we applied Gene Ontology (GO) enrichment analysis. Compared to tumor tissue, proteins belonging to lipid, amino acid, and glucose metabolic processes, and oxidative respiration were upregulated in TANM, likely due to the inclusion of surrounding liver tissue (Fig. 2c and Supplementary Fig. 2). In part, this observation also adheres to the commonly reported Warburg effect within solid tumors[39]. Conversely, proteins connected to the extracellular matrix (ECM), as well as proteins involved in DNA replication, and RNA transport and processing were upregulated in ICC (Fig. 2c and Supplementary Fig. 2). Enrichment of endocytosis, spliceosome, and DNA repair pathways, together with down-regulation of proteasome components, was previously described by Dong et al. as a trans-effect of a characteristic chromosome 14q loss[28]. We identified ECM and matrisome components (e.g., prolargin, mimecan, biglycan, lumican, and various collagens) to be among the most prominently increased proteins within ICC tumor tissue, which is in agreement with strong desmoplastic reactions being a hallmark of ICC[24]. Further, we surveyed the proteome data for proteins likely related to immune cell infiltration or tumor immunity (Fig. 2b and Supplementary Fig. 1B). We found pan-leukocyte marker receptor-type tyrosine-protein phosphatase C (PTPRC, also known as CD45) and neutrophil-specific marker CEA-CAM8 to be increased in ICC[40,41]. In contrast, the monocyte/macrophage markers CD14 and CD163 were significantly increased in TANM tissue, which might be related to their high expression in Kupffer cells of the liver (Fig. 2c and Supplementary Fig. 1B)[42–44].

### ICC tumor proteomes form two clusters with strong proteomic differences and divergent times to recurrence in the MSKCC-ICC cohort
Recent proteome studies on solid tumors, including ICC, have highlighted the existence of proteomic subgroups within a tumor entity; i.e., subgroups that are enriched in proteins stemming from (tumor-) biological processes[28,30]. Unsupervised PCA of our ICC protein abundance data of the MSKCC-ICC cohort showed homogenous distribution of tumor samples along components 1 and 2 (Fig. 3b). Next, we performed unsupervised hierarchical clustering, which sorted tumors into two proteomic clusters (Supplementary Table 1 and Fig. 3a). The number of clusters was confirmed using a Monte-Carlo-based consensus clustering analysis with hierarchical clustering set as method (Supplementary Fig. 3A). Linear modeling of differentially expressed proteins followed by gene set enrichment analysis (GSEA) via the KEGG and Gene Ontology databases revealed different biological processes to be upregulated in each cluster. In cluster 1, we detected a fingerprint of matrisomal proteins, humoral immune response, and coagulation and complement cascades. Cluster 2 exhibited increased activity in DNA replication and synthesis, and processing of (m)RNA and proteins (Fig. 3c, d, Supplementary Fig. 4, Supplementary Data 3). In line with their proteomic motifs, we refer to cluster 1 as "ECM cluster" and to cluster 2 as "proliferation cluster". Consistently, the routinely used proliferation markers KI67 and PCNA were both upregulated in the proliferation cluster (KI67 logFC = 0.65, $p_{adj} = 4.48E^{-4}$; PCNA logFC = 0.60, $p_{adj} = 1.81E^{-4})^{45}$. We further investigated the matrisomal proteins identified in the ECM cluster and found elevated expression to be largely confined to classical ECM glycoproteins, collagens, and proteoglycans (Supplementary Fig. 5). A screen for immunity-related marker proteins did not reveal specific inflammation patterns associated with either cluster (Supplementary Fig. 3B, C).

Next, we probed whether both clusters differ in clinicopathological features. Most clinical parameters, including sex, age, and duct type classification (large duct type and small duct type), were

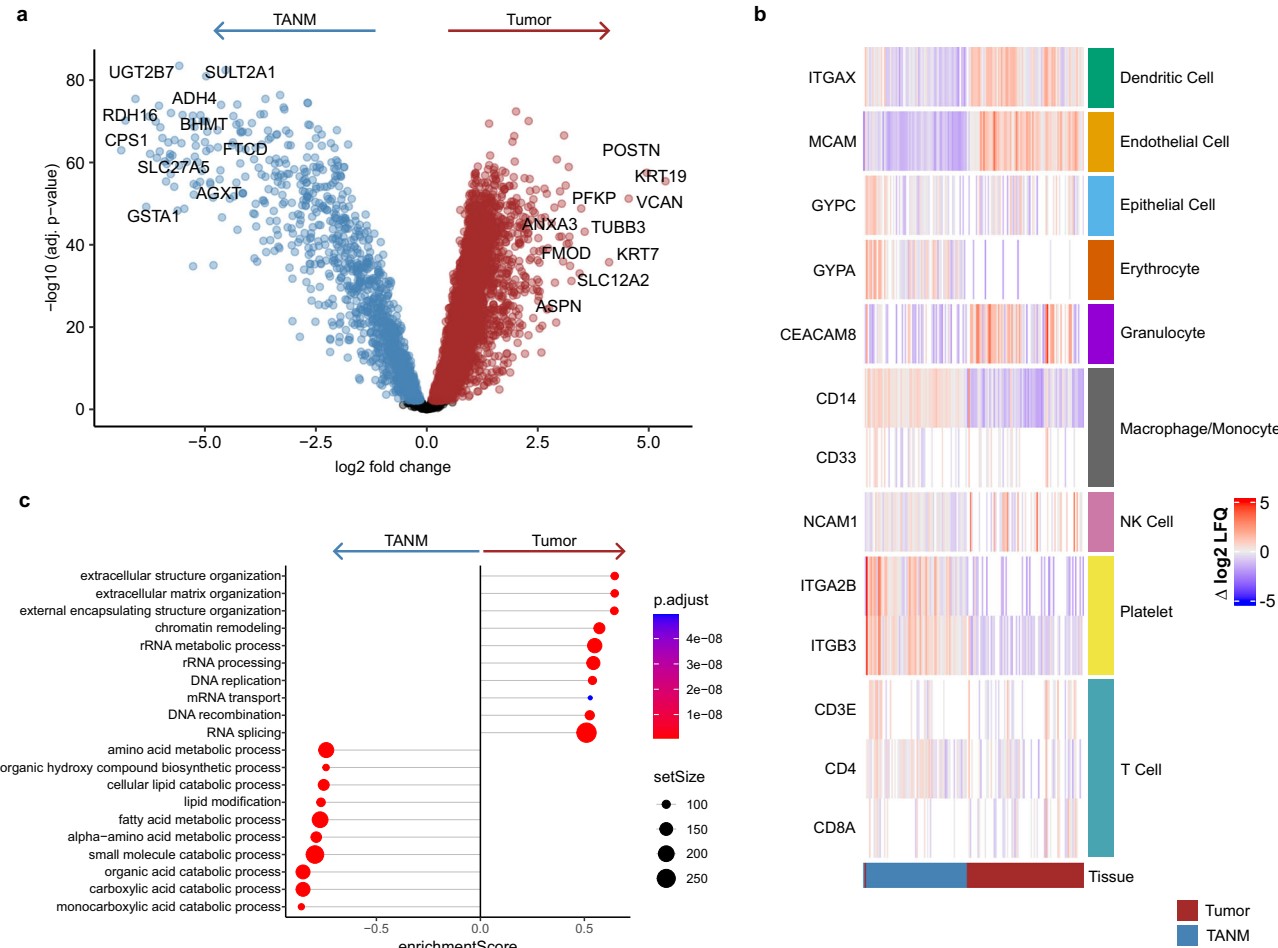

**Fig. 2 | Tumor vs. TANM tissue. a** Volcano plot of differentially expressed proteins between tumor ($n = 80$) and TANM ($n = 69$) tissues. Log-fold changes and moderated two-sided t-tests were obtained using limma linear models with Benjamini–Hochberg FDR correction. Dots colored in red (tumor) or blue (TANM) lie above the adjusted significance threshold. **b** Heatmap of immune cell marker proteins. The red bar indicates the tumor, the blue bar indicates the TANM tissue. The color scale represents row-wise median-normalized log2 intensities. **c** Gene Ontology gene set enrichment analysis (GSEA) of differentially regulated biological processes between tumor and TANM. The top 20 terms are shown. Significance of GSEA items was determined by a permutation test with false-discovery rate (FDR)-based multiple testing correction. Color indicates adjusted *p*-value, and point size indicates set size. Source data are provided as a Source Data file.

evenly distributed across both clusters (Supplementary Table 1, $p_{sex} = 0.4$ Pearson's Chi-squared test, $p_{age} > 0.9$ Wilcoxon rank sum test, $p_{ducttype} = 0.9$ Pearson's Chi-squared test). However, tumor stages differed significantly across clusters ($p = 0.014$): the proliferation cluster contained a higher fraction of AJCC stage T2 specimens, while the ECM cluster included more T1 tumors. Analysis of corresponding tissue images by a trained pathologist revealed that tumors of the proliferation cluster were characterized by significantly increased lymphangioinvasion and a tendency for elevated peritumoral tumor budding and satellite nodules, suggesting more invasive growth (Pearson's Chi-squared Test, $p_{lymphangio} < 0.001$, $p_{peri\_tubudding} = 0.065$, $p_{satellite\_nodule} = 0.083$). Most importantly, the ECM cluster is associated with significantly prolonged time-to-recurrence (TTR) compared with the proliferation cluster (median recurrence-free survival: proliferation 229 days vs. ECM 859 days, logrank $p = 0.00013$, total recurrence events 27/48 ECM and 29/32 proliferation cluster patients, Fig. 3e). The clusters are more predictive for recurrence-free survival than AJCC tumor staging (Supplementary Fig. 6A). Moreover, both clusters are significant predictors for overall survival (OS) (median survival: proliferation 872 days vs. ECM 1817 days, logrank $p = 0.014$, total events 19/48 ECM and 20/32 proliferation cluster patients, Supplementary Figs. 6B and 7A, C).

30 out of 80 MSKCC-ICC patients have been treated with adjuvant radiotherapy, with a significantly larger fraction belonging to the proliferation cluster (proliferation: 18/32 vs ECM: 12/48; Chi-Squared Test $p_{Radiotherapy} = 0.005$, Supplementary Table 1). Still, radiation therapy remains less predictive for recurrence-free survival than the proteome clusters and was linked to significantly shorter TTRs (Fig. 4a, b and Supplementary Fig. 6A).

Furthermore, we used an image-based classification model to deconvolute different cell types present in tumor samples. The classifier was trained in a supervised manner with more than 150,000 manually annotated nuclei by a surgical pathology resident. Annotated cell categories included immune cells, liver cells, necrotic cells, stromal cells and tumor cells. We found no difference between the two clusters regarding liver and immune cell content, whereas necrotic and stromal cells were significantly enriched in tumors belonging to the ECM cluster. Interestingly, there was no difference in the amount of tumor cells, underlining that the two clusters reflect distinct tumor phenotypes rather than differences in tumor cell content (Pearson's Chi-Square test, $p_{Liver\ Cells} = 0.2$, $p_{Immune\ Cells} = 0.3$, $p_{Necrotic\ Cells} = 0.005$, $p_{Stromal\ Cells} = 0.013$, $p_{Tumor\ Cells} = 0.15$, Supplementary Table 1). Importantly, tumor cellularity was not associated with either TTR or OS

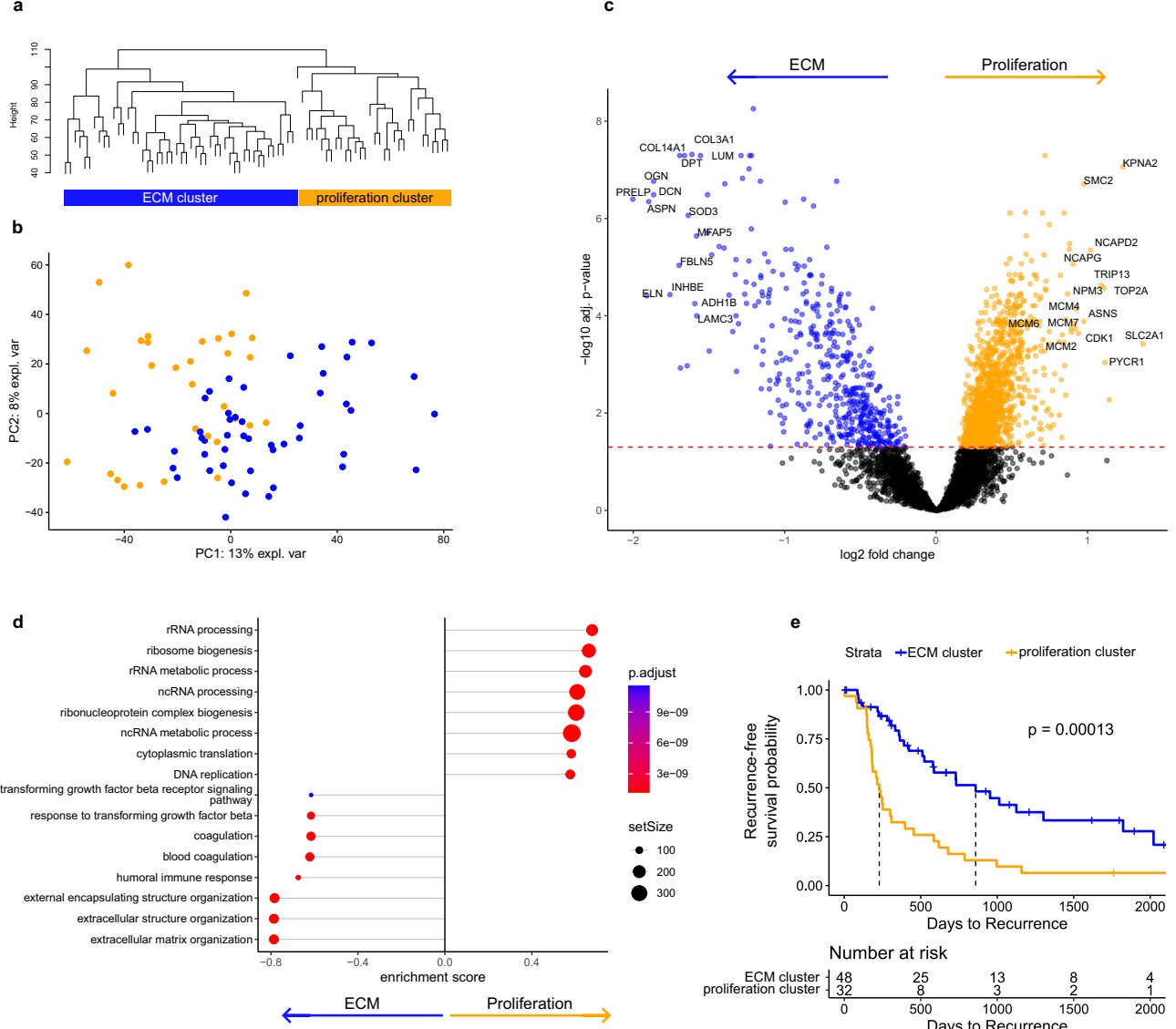

**Fig. 3 | Hierarchical clustering. a** Hierarchical clustering of tumor samples. **b** Principal component analysis (PCA). Dots colored in orange represent the proliferation cluster, and dots colored in blue represent the ECM cluster. **c** Volcano plot of differentially expressed proteins. Log-fold changes and moderated two-sided *t*-tests were obtained using limma linear models with Benjamini–Hochberg false discovery rate correction. Dots colored in orange (proliferation cluster; $n = 32$) or blue (ECM cluster; $n = 48$) lie above the Benjamini-Hochberg-adjusted significance threshold. **d** Gene Ontology (Biological Process) gene set enrichment analysis (GSEA) of differentially regulated biological processes between both clusters. The top 16 terms are shown. Significance of GSEA items was determined by a permutation test implemented in clusterProfiler with Benjamini–Hochberg-based multiple testing correction. Color indicates adjusted *p*-value, and point size indicates set size. **e** Kaplan–Meier curve incl. log-rank test comparing TTR distribution between ECM (in blue) and proliferation cluster (in orange). Source data are provided as a Source Data file.

(Cox proportional hazards model, $HR_{TTR} = 1.008$, $p_{TTR} = 0.356$, $HR_{OS} = 0.995$, $p_{OS} = 0.656$; Supplementary Fig. 6C).

These proteomic results are largely in line with previous multi-omics publications. Dong et al. describe four ICC subclusters based on mRNA, proteomic, and phosphoproteomic data for the FU-ICC cohort[28]. FU-ICC cluster 1 was enriched in proteins involved in inflammation and glucose metabolism. Conversely, FU-ICC cluster 2 showed high levels of ECM proteins. FU-ICC cluster 3 was enriched with proteins involved in MAP kinase signaling and fatty acid and nucleotide synthesis, while the biology of FU-ICC cluster 4 remained inconclusive. FU-ICC cluster 1 had the worst prognosis, particularly for tumors of AJCC stages I and II. Similarly, Lin et al. report three immune subgroups within the same cohort and link immune-activated cluster IG3 (fibroblast and lymphocyte invasion) to prolonged survival, while other clusters IG1 and IG2 are characterized by a worse prognosis and

innate immunity mediated inflammation (IG1) or immune exclusion (IG2)[29]. A study by Deng et al. was not fully comparable due to the inclusion of extrahepatic cholangiocarcinoma, which were characterized by a very high expression of matrisomal proteins[30]. Interestingly, Martin-Serrano et al. described contrasting inflamed vs. non-inflamed ICC gene expression profiles in a large re-analysis of several transcriptomic datasets. Inflamed tumors, representing about 40% of patients, were defined by strong stromal enrichment but not clearly associated with survival outcome[24].

## Cox proportional hazards model highlights individual prognostic ICC proteins in the MSKCC-ICC dataset

To identify biological motifs and individual proteins with expression patterns directly correlating to the TTR, we iteratively applied a Cox proportional hazards model (CPHM) within the MSKCC-ICC cohort.

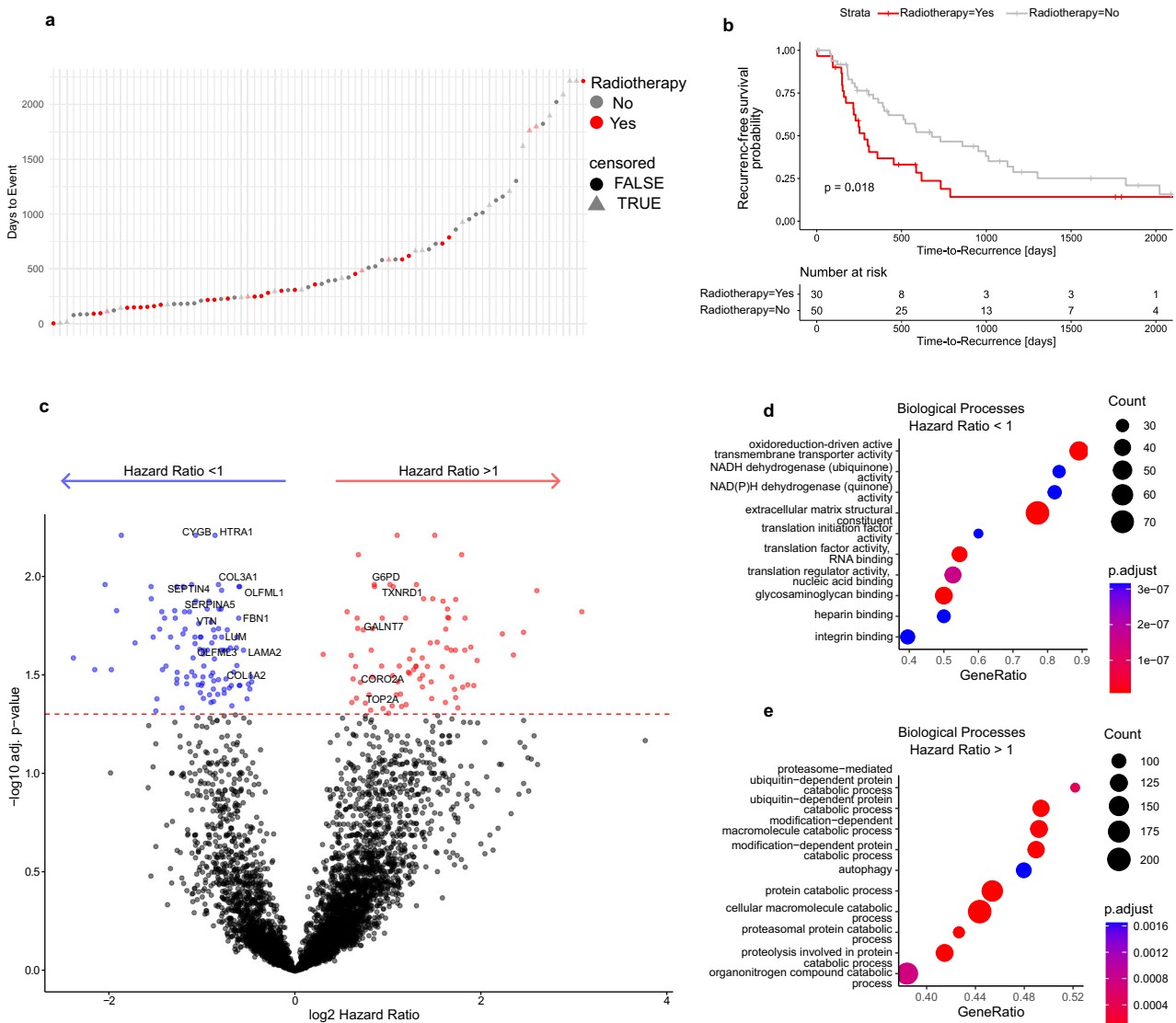

**Fig. 4 | Survival statistics. a** Time-to-recurrence (TTR) distribution across the cohort. Red indicates patients who received radiotherapy, and gray indicates patients who did not. Point shape denotes censoring status (non-censored as circles; censored as triangles). Time-to-Recurrence (TTR) distribution across the cohort. Red color indicates radiotherapy, grey color indicates no radiotherapy. Shape highlights censoring (non-censored in round shape, censored in triangular shape). **b** Kaplan–Meier curve incl. two-sided log-rank test comparing patients with radiotherapy (in red) to patients without (in gray). **c** Volcano plot of Cox proportional hazards model (CPHM) results for protein abundance with radiotherapy included as covariable. Log2-transformed hazard ratios and Benjamini-Hochberg adjusted, two-sided *p*-values are shown. Dots colored in red (Hazard Ratio > 1) or blue (Hazard Ratio < 1) lie above the adjusted significance threshold. **d**, **e** Gene Ontology gene set enrichment analyses (GSEA) of upregulated biological processes among proteins with high and low hazard ratios. Top ten terms are shown. Significance of GSEA items was determined by a permutation test with false-discovery rate (FDR)-based multiple testing correction. Color indicates adjusted *p*-value, and point size indicates set size. Source data are provided as a Source Data file.

Adjuvant radiotherapy was included as an independent covariate, since it was significantly associated with a shorter TTR (Fig. 4a, b). In total, we identified 188 proteins with expression patterns significantly correlating with the TTR distribution (*p*.adj < 0.05, Fig. 4c). Among these, 84 proteins were associated with an increased risk of early recurrence (hazard ratio (HR) > 1), while 104 proteins were associated with a reduced risk (HR < 1). Gene set enrichment analysis revealed that proteins with high HRs were enriched in pathways related to proteasomal degradation and autophagy (Fig. 4e). Conversely, proteins with HR < 1 were involved in ECM and oxidoreductive processes (Fig. 4d). Taken together, the biological processes associated with high or low recurrence risk were in line with those revealed by hierarchical clustering.

Importantly, CPHM does not consider absolute protein abundance and thus also recognizes subtle expression changes as long as they align with the TTR distribution. To highlight protein hits with meaningful abundance changes across the cohort, we further filtered CPHM results according to their log2 abundance coefficient of variation. Only proteins with a standard deviation >1 were considered, yielding 44 proteins. Of these, 5 proteins with high HR were also significantly enriched (logFC > 0.8, *p*.adj < 0.05) in the unfavorable proliferation cluster—for example, DNA topoisomerase 2-alpha (TOP2A) and glucose-6-phosphate 1-dehydrogenase (G6PD). Conversely, 28 proteins were significantly enriched in the favorable ECM cluster (logFC > 0.8, p.adj <0.05), including prolargin (PRELP), mimecan (OGN), vitronectin (VTN), and lumican (LUM; Supplementary Data 4). Several of the identified high-risk candidate proteins have been previously associated with the tumor biology of cholangiocarcinoma. For example, high TOP2A expression is associated with a poor prognosis[46]. Similarly, G6PD overexpression has been reported for several gastrointestinal cancers and is also related to clinical outcome[47]. Conversely,

several of the proteins with low HR and enriched in the ECM cluster have been previously associated with tumor suppression and/or better prognosis in other gastrointestinal cancer types, such as cytoglobin, serine protease HTRA1 and septin-4[48–51].

Using the same CPHM workflow, we also identified 74 proteins correlating with OS (Supplementary Data 4). Of these proteins, 46 overlapped with the TTR CPHM analysis; however, they showed slightly diverging biological motifs in a GSEA (Supplementary Fig. 7D). Here, proteins involved in protein degradation imply a higher risk, whereas proteins linked to intracellular transport correlate with prolonged OS.

## A classifier for ICC clusters detects similar proteomic motifs and survival outcome in the FU-ICC cohort

To transfer the concept of "prognostic ICC subtypes" to new and/or third-party proteomic ICC data, we trained a machine-learning-based classification model. From the 50 most cluster-defining proteins as determined by log2 fold-change and adjusted $p$-value in the differential expression analysis, we randomly generated 100 combinations of four proteins. Each of these classifier-candidate sets contained two proteins upregulated in the ECM cluster and two proteins upregulated in the proliferation cluster. Notably, this comprised 24 of the 33 proteins we determined as particularly related both to the cluster identities and to the TTR. For each set, we trained a classification model by partitioning the MSKCC-ICC dataset (tumor samples only) into a training and a test set (75% and 25% of samples, respectively; Fig. 5a). Across all 100 tested models, we obtained receiver operating characteristic (ROC) curves with a mean area under the curve (AUC) of 0.89; with only five sets falling slightly below an AUC of 0.8. Mean sensitivity and mean specificity were 0.86 and 0.88, respectively (Fig. 5b). These results emphasize that a broad range of the most differentially abundant proteins between both clusters, and combinations thereof, are well suited for building classifier models.

To validate our classifier approach and the clinical relevance of the clusters, we applied the model to all 214 cases with proteomic data from the FU-ICC dataset published by Dong et al.[28], after having adjusted it to the MSKCC-ICC intensity range using the ComBat algorithm. As a classifier, we selected the set of proteins with the highest log2 fold-change and lowest adjusted $p$-value in the MSKCC-ICC cohort. All four proteins—prolargin (PRELP), mimecan (OGN), structural maintenance of chromosomes protein 2 (SMC2), and importin subunit alpha-1 (KPNA2)—were also consistently identified (<20% missing values) across the FU-ICC dataset. For a more in-depth assessment, we additionally evaluated this classifier's performance across 100 bootstrapped MSKCC-ICC-based training and test datasets (random 75%/25% partitions), yielding an average AUC of 0.9. When applied to the FU-ICC cohort, the classifier categorized it into 141 ECM and 73 proliferation cluster cases (Fig. 5c, d; Supplementary Fig. 8B; Supplementary Table 3). Differential expression analysis between the assigned clusters in the FU-ICC dataset revealed biological patterns consistent with those observed in our MSKCC-ICC cohort (Supplementary Fig. 8A, C). Moreover, patients assigned to the ECM cluster demonstrated significantly prolonged overall survival as compared to those of the proliferation cluster (survival data available from 137 patients, median survival: ECM not reached vs. proliferation 650 days, logrank $p = 0.00066$, HR proliferation cluster = 2.1; Fig. 5e). Looking at the clinical data, tumors of the assigned proliferation cluster were more often categorized as TNM-stage II (Pearson's Chi-squared Test, $p_{TNM} = 0.007$) with significantly higher intrahepatic metastasis and vascular invasion (Pearson's Chi-squared Test, $p_{ihep\_met} = 0.007$, $p_{VI} = 0.022$; Supplementary Table 3).

The FU-ICC dataset also included genomic data. Interestingly, the assigned proliferation cluster contained a higher fraction of overall mutations, while the ECM cluster samples carried more (FGFR)-fusions (Supplementary Fig. 8D–F). FGFR2-fusions in ICC have recently been reported to be associated with a favorable diagnosis, which aligns with our findings[52]. Also, cancer-associated genes *TP53* and *MUC16/CA125* were more frequently mutated in the proliferation cluster (Fisher's Exact Test, $p_{TP53} = 0.0003$, $p_{MUC16} = 0.0002$, Supplementary Fig. 8F), reflecting the worse survival outcome Dong et al. reported for tumors harboring *TP53* mutations[28]. We observed a similar pattern in the RNA-seq data, with ECM-related genes upregulated in the proteomic ECM cluster and replication-related genes upregulated in the proteomic proliferation cluster (Supplementary Fig. 8G, H). Unfortunately, similar genomic studies were not feasible for the MSKCC-ICC cohort owing to a lack of suitable sample material.

## Classification of the UKF-ICC validation cohort reveals matching prognostic clusters

Encouraged by these results, we integrated an additional ICC cohort consisting of samples collected at the University Medical Center Freiburg (UKF-ICC) as a second validation dataset. This cohort included matching tumor and TANM tissues from 66 patients provided as FFPE cores. Proteomic measurement was performed on a Thermo Q Exactive Plus, yielding an average of 4540 identified proteins (3060 proteins with less than 20% missing values), following the data analysis strategy described above (Supplementary Fig. 9A; Supplementary Data 5). After consultation with a pathologist, four tumor samples (plus the matched tissue) were excluded due to ambiguous classification, resulting in 62 tumor samples with 61 patient-matched TANM samples. Again, tumor and TANM tissues separated well in clustering analyses (Supplementary Fig. 9C). A total of 2581 proteins were differentially abundant between tumor and TANM tissue ($p < 0.01$, FDR-corrected, Supplementary Fig. 9B). Findings were very similar to the MSKCC-ICC cohort, with established ICC markers such as KRT7 and KRT19 being more abundant in tumors (KRT7 logFC = 4.21, $p_{adj} = 1.83E^{-27}$; KRT19 logFC = 4.31, $p_{adj} = 1.12E^{-36}$). In contrast, TANM tissue showed higher abundance of common liver enzymes, such as aldolase B (ALDOB), alcohol dehydrogenases and arginase 1 (ARG1), confirming the accuracy of pathological sampling (ALDOB logFC = −6.22, $p_{adj} = 5.14E^{-48}$, ADH1A logFC = −5.93, $p_{adj} = 2.22E^{-43}$, ARG1 logFC = −4.95, $p_{adj} = 1.82E^{-51}$). Gene set enrichment analysis revealed upregulation of RNA processing and downregulation of metabolic and catabolic processes in ICC tissue compared to TANM (Supplementary Fig. 9D).

Using the same approach as for the FU-ICC cohort, we classified tumor samples into the ECM and proliferation cluster by using a 4-protein classifier. Again, proteins were selected for consistent identification in both cohorts, as well as their log2-fold changes and adjusted $p$-values in the MSKCC-cohort. For the UKF-ICC cohort, the classifier set consisted of prolargin (PRELP), mimecan (OGN), DnaJ homolog subfamily C member 7 (DNAJC7), and nucleolar RNA helicase 2 (DDX21). As for the FU-ICC cohort, we evaluated the classifier's performance across 100 bootstrapped MSKCC-ICC-based training and test datasets, yielding an average AUC of 0.89 (Fig. 5g). The cohort was stratified into 18 ECM and 42 proliferation cluster samples (2 samples were excluded due to incomplete clinical annotation; Fig. 5h; Supplementary Fig. 10B; Supplementary Table 2). Again, the ECM cluster was associated with significantly longer overall survival (median survival: ECM 46 vs. proliferation 25 months, log-rank $p = 0.029$, Fig. 5i). We also observed trends towards more grade 2 tumors and increased perineural sheath invasion in the proliferation cluster (Pearson's Chi-squared Test, $p_{TNM} = 0.076$, $p_{PNI} = 0.073$, Supplementary Table 2). 49 proteins were differentially abundant between the two clusters, and a GSEA revealed upregulation of glycerophospholipids in the ECM cluster and upregulation of translation in the proliferation cluster (Supplementary Fig. 10A, C). Compared to the other cohorts, protein expression differences between clusters were less pronounced, likely reflecting the use of FFPE tissue punches that can contain minor amounts of surrounding tissue, diluting proteomic specificity. Whole-exome sequencing (WES) of 19 tumors and matched TANM samples

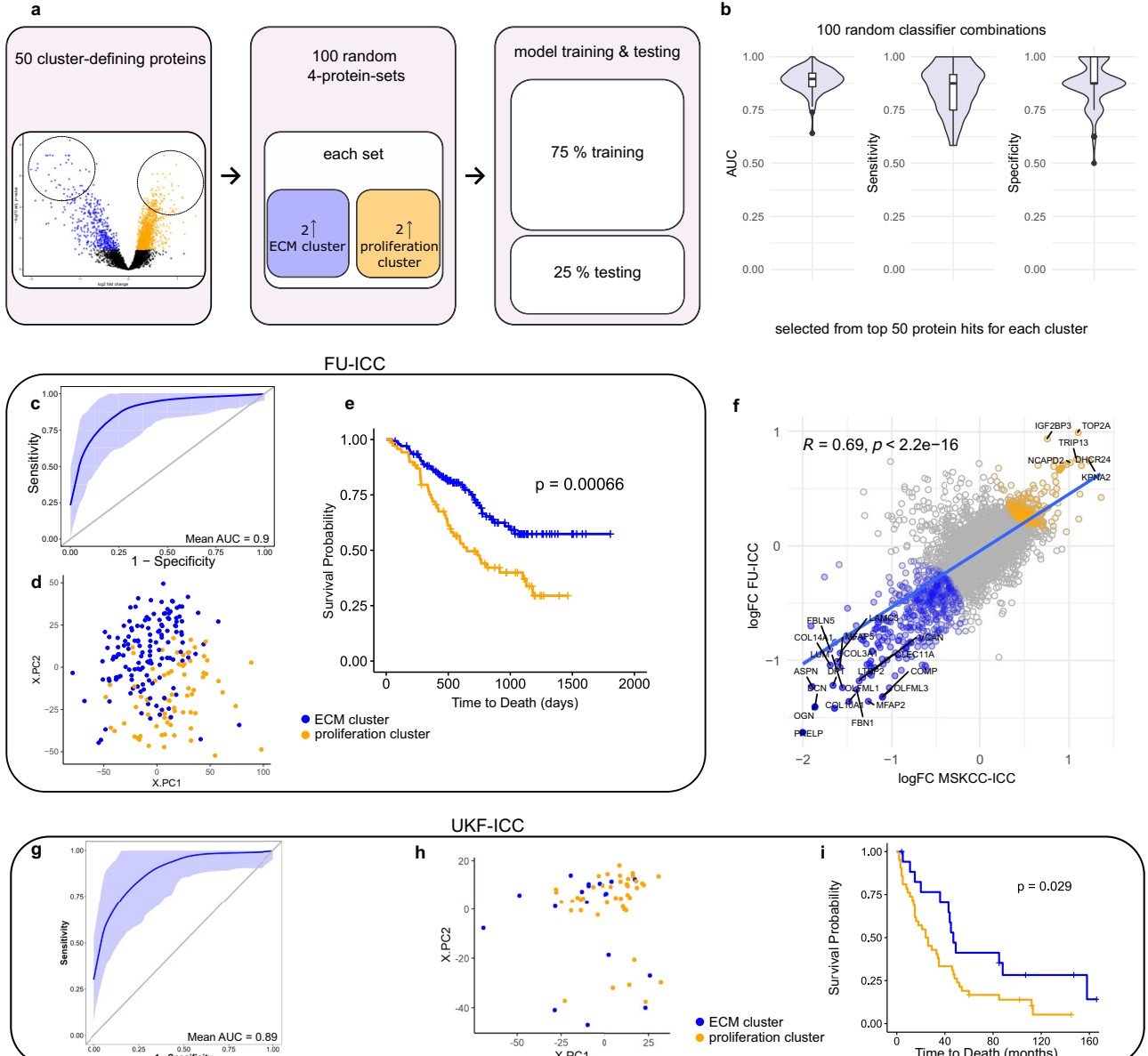

**Fig. 5 | Classification model. a** Schematic overview of classifier generation. **b** Violin plots of AUC, sensitivity, and specificity across 100 randomly selected four-protein classifier combinations. Boxplots show median (center line), interquartile range (IQR, extending from the 1st to the 3rd quartile, box), and 1.5 IQR (whiskers). **c** 100-fold bootstrapped ROC curves of the classifier as trained and tested on tumor samples in the MSKCC-ICC dataset, which was iteratively split into randomized training (75%) and test (25%) sets. Shaded regions represent 95% bootstrap confidence intervals around the mean ROC curves. **d** PCA of FU-ICC cohort indicating assigned clusters (ECM in blue; proliferation in orange). **e** Kaplan–Meier curve incl. log-rank test comparing overall survival in the FU-ICC cohort between classifier-assigned ECM ($n$ = 141) and proliferation ($n$ = 73) clusters. Dotted lines in Kaplan-Meier curves indicate median survival (ECM in blue; proliferation in orange).

**f** Correlation of protein logFC values between the MSKCC and FU ICC cohorts (ECM vs. proliferation clusters). Colors indicate proteins with concordant enrichment in the ECM (blue) or proliferation (orange) cluster across both cohorts. **g** 100-fold bootstrapped ROC curves of the applied classifier for the UKF-ICC cohort, as trained and tested on tumor samples in the MSKCC-ICC dataset, which was iteratively split into randomized training (75%) and test (25%) sets. Shaded regions represent 95% bootstrap confidence intervals around the mean ROC curves. **h** PCA of UKF-ICC cohort indicating assigned clusters (ECM in blue; proliferation in orange). **i** Kaplan–Meier curve incl. log-rank test comparing overall survival in the UKF-ICC cohort between classifier-assigned ECM ($n$ = 18) and proliferation ($n$ = 42) clusters. Dotted lines in Kaplan-Meier curves indicate median survival (ECM in blue; proliferation in orange). Source data are provided as a Source Data file.

identified 512 somatic mutations, with a low overall mutational burden (mean TMB = 0.73 mut/mb; Supplementary Data 6). Due to limited sample size and mutation frequency, the genomic data remained inconclusive (Supplementary Fig. 10D).

## Semi-specific peptides are increased in TANM and differ between clusters

Tumors and cells of the tumor microenvironment, such as immune cells, actively modify surrounding ECM through endogenous proteolysis, which may aid tumor progression and metastasis[53]. This process yields truncated proteins that are amenable to proteomic analysis[54]. Recently, we have shown that so-called semi-specific approaches in peptide-to-spectrum matching are a valuable approach to grasping endogenous proteolytic processing[55,56]. Since all samples of the present study were digested with trypsin, which specifically cleaves protein sequences after arginine or lysine, any detected peptide sequence that deviates from this specificity might represent a (patho-)

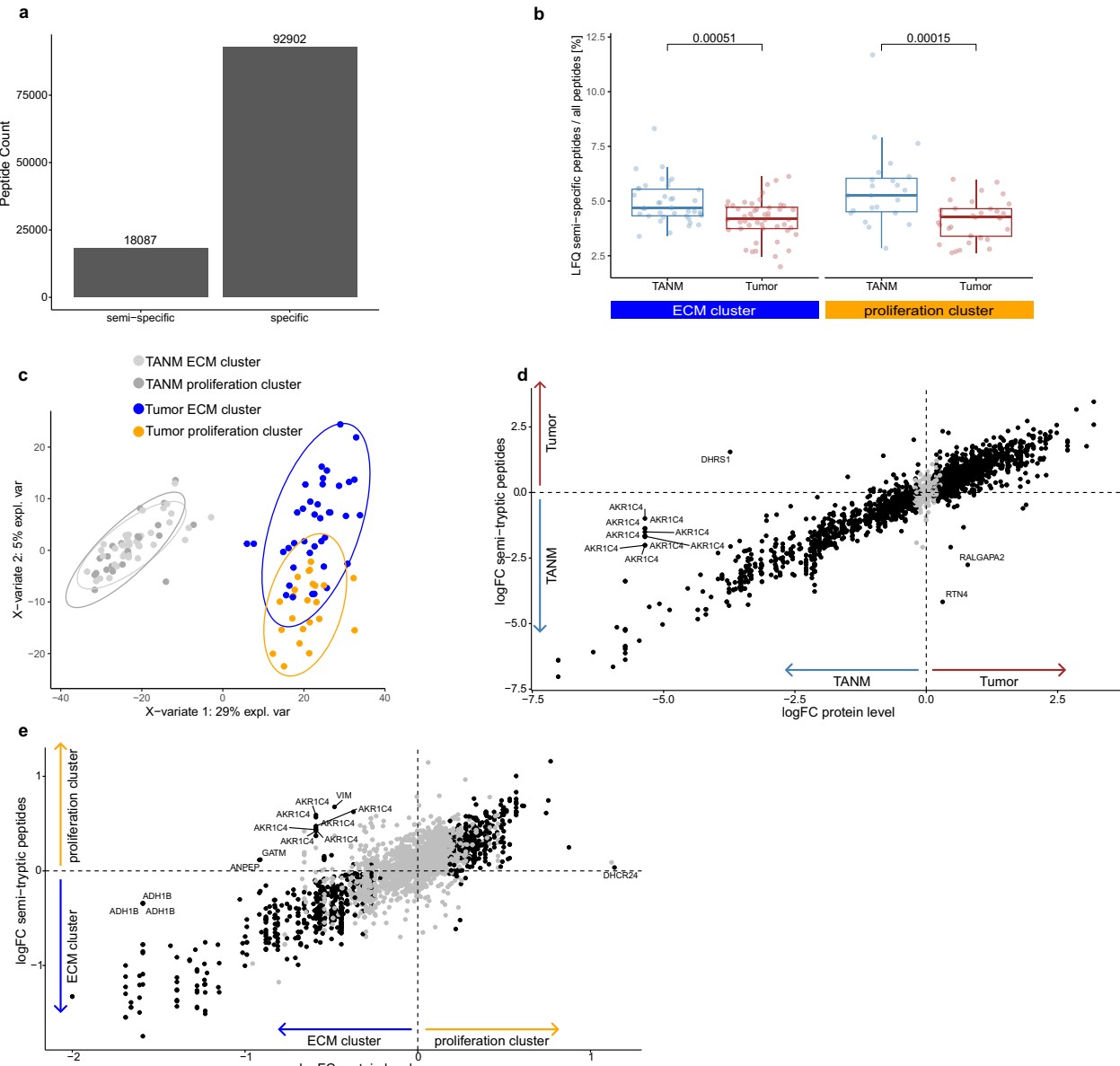

**Fig. 6 | Semi-tryptic analysis. a** Count of fully tryptic (specific) and semi-specific detected peptides. **b** Boxplot indicating the ratio of semi-specific peptides among all detected peptides for tumor (in red) and TANM (in blue) sample pairs in the ECM (*n* = 39) and proliferation cluster (*n* = 23). Unpaired, two-sided *t*-test. Boxplots show median (center line), interquartile range (IQR, extending from the 1st to the 3rd quartile, box), and 1.5 IQR (whiskers). **c** Partial Least Squares-Discriminant Analysis (PLS-DA) of semi-specific peptide expression in both clusters (ECM in blue, proliferation in orange) and matching TANM tissues (ECM in light gray, proliferation in dark gray). Ovals indicate 95% confidence intervals. **d** Correlation of logFC values between protein abundance and semi-tryptic peptide abundance in tumor vs. TANM tissue. **e** Correlation of logFC values between protein abundance and semi-tryptic peptide abundance in ECM vs. proliferation clusters. Source data are provided as a Source Data file.

physiological cleavage product. Across the MSKCC-ICC cohort, 16.3% (18,087) of all identified peptides possessed such semi-tryptic properties (Fig. 6a). When looking at the different tissues, we find significantly higher portions of semi-specific peptides in TANM samples, regardless of the matching tumor cluster (unpaired *t*-tests, $p_{ECM}$ = 0.000511, $p_{proliferation}$ = 0.00015, Fig. 6b). Interestingly, semi-specific peptides seem to differ between tumor and TANM tissue—and also among both clusters (Fig. 6c). While the abundance of most semi-specific peptides closely correlates with their protein of origin, few semi-specific peptides show a diverging behavior. For example, although dehydrogenase/reductase SDR family member 1 (DHRS1) was decreased in tumor tissues, one semi-specific DHRS1 peptide was significantly enriched (Fig. 6d). Similarly, multiple semi-specific

peptides of the functionally related aldo-keto reductase family 1 member C4 (AKR1C4) did not correlate well with whole protein abundance when compared between tissues and clusters (Fig. 6e). We also found several proteases to be differentially abundant between the different tissues and clusters (Supplementary Fig. 11). Additionally, we considered semi-specific peptides in the UKF-ICC cohort (Supplementary Fig. 12A). Consistent with our main findings, the proportion of semi-specific peptides was higher in TANM compared to tumor samples in both clusters (Supplementary Fig. 12B). We also observed substantial global differences in semi-specific peptide abundances between TANM and tumor tissue, whereas differences between clusters were minimal (Supplementary Fig. 12C–E), possibly due to lower purity of FFPE punch samples and/or higher sparsity in the semi-specific

dataset. Interestingly, RTN4 emerged as the only protein with a peptide not correlating with the abundance of its parent protein—consistent with its behaviour in the MSKCC-ICC cohort (Fig. 6D and Supplementary Fig. 12D).

Recently, it has been demonstrated that antibodies may specifically recognize proteolytically generated neo-epitopes in solid tumors[57]. Identification of tumor-specific patterns of endogenous proteolytic processing may supplement such efforts.

### Proteomics of ICC PDX models enables insight into tumor-stroma co-regulation

Since ICC is characterized by strong desmoplastic reactions, the differences in ECM patterns and recurrence-free survival between clusters highlight the need to better understand tumor-stroma co-regulation[24]. Patient-derived xenograft (PDX) mouse models offer unique opportunities for studying tumor-stroma interactions[58]. Given that amino acid sequences between murine and human proteins are different, the species origin of individual proteins can be inferred from MS-based proteomic data[59].

In this study, we investigated the interplay of ICC tumors and their surrounding stroma on the proteome level in a small cohort of nine PDX, obtained from a commercial source independent of the patient cohorts. Since the transplanted tumor was human, invading stromal cells were supposed to be of murine origin. To unambiguously assign human and murine proteins to our DIA data, we only considered peptides with sequences that are unique to the mouse or human proteome. Data on PDX FFPE specimens were acquired on a Thermo Q Exactive Plus, leading to more than 4600 identified proteins per sample. On average, we detected 1491 murine (stroma) and 3168 human (tumor) proteins per sample (Fig. 7a and Supplementary Data 7). These numbers underscore the feasibility of performing mixed-species proteomics experiments with an acceptable proteome coverage.

We generated an unsupervised partial least squares (PLS) model of the data, defining co- or inversely regulated sets of proteins independently of their species-origin. For the first two components, we focused on the top 200 proteins with the highest absolute factor loadings. With this approach, we could identify proteins sharing similar expression patterns throughout the PDX cohort and functionally compare them to their inversely regulated counterparts. The coregulation is illustrated by a circle plot (Fig. 7b). In brief, proteins aggregating together have similar expression patterns, whereas proteins on opposing ends of either axis are inversely regulated.

Murine and human proteins form contrasting sets along component 1, meaning that the strongest inverse expression pattern is determined by species origin. An enrichment analysis of the negative axis end of component 1 mainly showed murine proteins associated with neutrophil degranulation, ECM organization, cellular respiration, as well as glucose and amino acid metabolism. At the opposing, positive axis end, we found human proteins involved in mitochondrial and cytosolic translation, ribosome biogenesis, and protein metabolism. This divergent expression pattern reflects clustering motifs within the human ICC cohort and supports the notion that innate immune reactions and ECM deposition are initiated by stromal cells invading the tumor. Accordingly, we observe a similar contrast in protein expression along component 2. At the positive axis end, human proteins were enriched for oxidative stress response and metabolism (mainly of lipids). At the negative axis end, murine proteins linked to ECM organization and collagen synthesis were increased. Regarding the interpretation of results, it needs to be considered that xenograft-carrying mice are deficient in adaptive immunity and thus cannot mount T-cell-mediated antitumor responses. Still, many detected murine ECM components were homologues to ECM cluster-associated proteins in the human cohorts. Since all used mice were isogenic, genomic diversity was confined to grafted tumors. Thus, as observed in our cohort and these xenografts, different translational and metabolic activities of the tumors might result in differing stromal responses.

### The mTOR effector EIF4A1 is strongly enriched in tumors of both clusters and its inhibition reduces tumor growth in ICC PDX

In recent years, the translation initiation factor EIF4A1 has gained interest as a downstream effector molecule of mTOR and a novel therapeutic target in various cancers. EIF4A1 plays a critical role in initiating the translation of selected mRNAs, including some that promote cancer, and is often overexpressed in proliferating cells[60,61]. For instance, elevated levels of EIF4A1 have been observed in hepatocellular carcinoma and pancreatic ductal adenocarcinoma[62,63]. Inhibiting EIF4A1 in cell lines from these cancers led to reduced proliferation and decreased oncogenic signaling[61–63]. Currently, a novel EIF4A1 inhibitor eFT226 (Zotatifin, MedChemExpress, NJ, USA) is being evaluated in two phase I/II clinical trials for breast cancer (clinicaltrials.gov NCT05101564 and NCT04092673). A recent study shows transcript-level upregulation of EIF4A1 in ICC and an anti-proliferative impact of Zotatifin in ICC cell models and organoids[64]. In our study, we noticed protein-level upregulation of EIF4A1 in ICC tumor tissues of both clusters (paired t-tests, adj. $p_{\text{ECM cluster}}$ = 9.9E-16, adj. $p_{\text{proliferation cluster}}$ = 1.3E-10, Fig. 8a). We performed an eFT226 titration of three ICC cell lines and found that, in two out of the three cases, eFT226 concentrations in the low nanomolar IC50 range reduced cellular viability (Supplementary Fig. 13), consistent with recently published data[64]. The addition of eFT226 to rapamycin (200 nM) yielded an additional reduction of cellular viability. On the other hand, the addition of eFT226 to treatment with gemcitabine (1 μM for gemcitabine-sensitive SNU-1079, 10 μM for HuCCT-1 and HuH-28) had only a minor effect on viability. Next, we aimed to investigate the compound in a more physiological setting that includes stromal reactions. In accordance with previous publications on eFT226, we conducted a study using ICC PDX in mice to evaluate the effect of 1 mg/kg eFT226 over an observation period of 35 days[65–67]. We used four mice each for the treatment and control groups, which were dosed with the drug or vehicle once per week for three weeks. We chose LI-002 (TW512) as a model, since stromal proteins were comparatively low-abundant here. Our findings show that eFT226 reduces tumor growth in PDX mice, leading to significantly lower tumor volumes (unpaired t-tests, $p_{\text{d14}}$ = 0.025, $p_{\text{d18}}$ = 0.006, $p_{\text{d21}}$ = 0.012, Fig. 8b; Supplementary Data 8) but not to the killing of tumor cells. Maximum anti-tumor efficacy occurred on study day 25, corresponding to moderate efficacy ($T/C_{\text{d25}}$ = 44.8%). Moreover, except for one animal with an ulceration, treated mice lived longer, tolerated the treatment well, and remained in good general condition throughout the experiment, as evidenced by the relative maintenance of their body weight (unpaired t-tests, $p_{\text{d18}}$ = 0.004, $p_{\text{d21}}$ = 0.002; Supplementary Data 4 and Supplementary Table 4; Fig. 8c). In contrast, three out of four mice in the control group had to be euthanized due to the tumor load until day 25 of the study (Supplementary Data 4 and Supplementary Table 4). This is in line with previous publications, which report growth inhibition or moderate regression, but no remission, in response to eFT226 treatment in other tumor entities[62,66]. Zhao et al. and Gerson-Gurwitz et al. also underscore that the effectiveness of eFT226 therapy can be greatly enhanced through synergistic treatment combinations[66,67].

In line with recent reports, our results encourage further translational research on eFT226 as a potential treatment for ICC patients[64]. Ongoing clinical studies in humans, eFT226's good tolerance in mice, and its high efficacy as mono- or combination treatment in models of other cancers underline its promising potential for ICC therapy. Overall, this successful translation of explorative proteomic data into new treatment rationales highlights the great potential of mass spectrometry-based approaches for translational research.

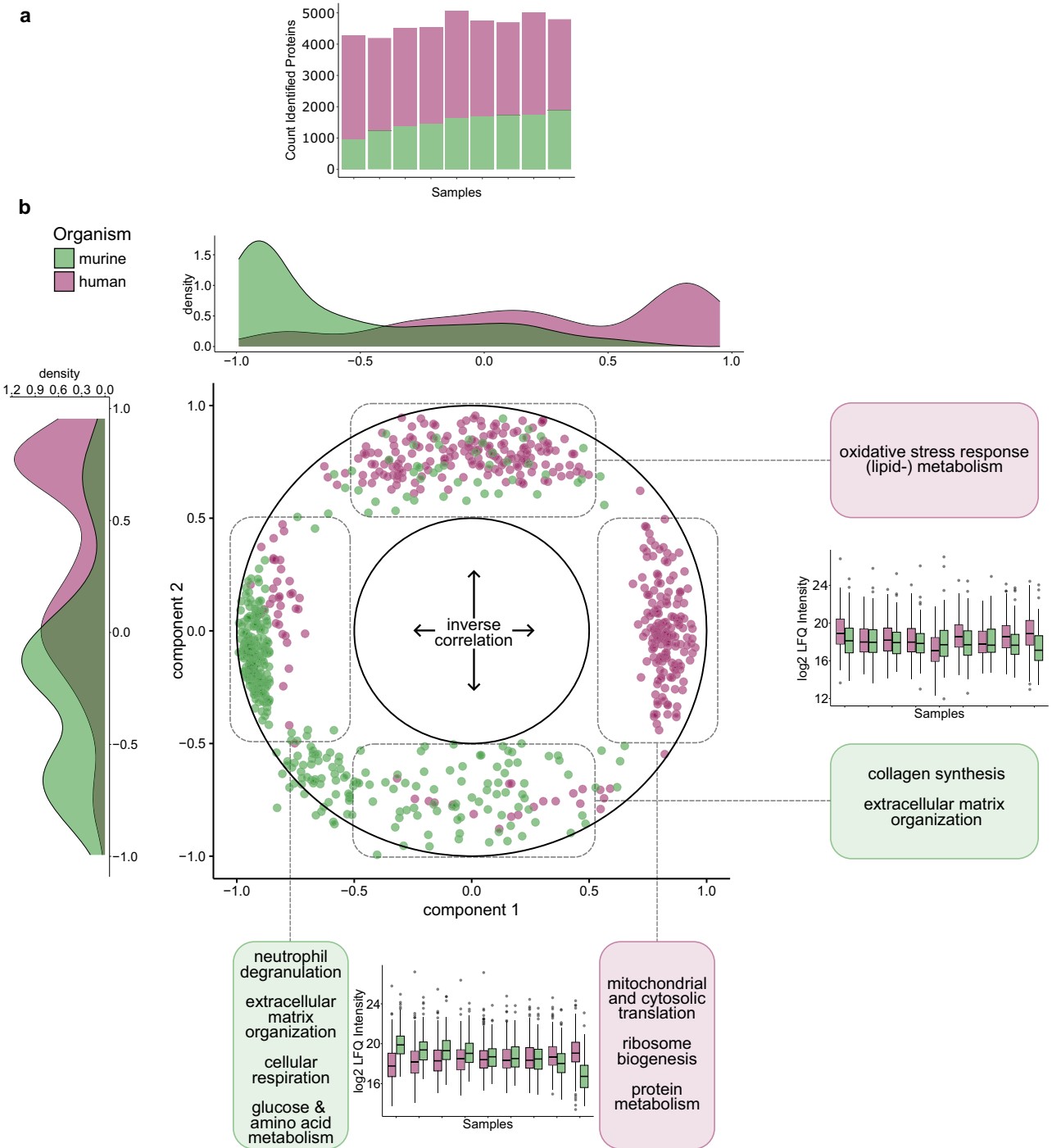

**Fig. 7 | Xenografts. a** Barplot of all detected protein groups per sample. Green color indicates murine origin, pink color indicates human origin. **b** Circle plot detailing an unsupervised Partial Least Squares (PLS) analysis of the top 200 proteins with the highest factor loadings. Proteins with similar expression patterns cluster together. Proteins with inverse expression patterns aggregate at opposing axis ends. Component 1 represents the strongest contrast in expression regulation, component 2 the second strongest. Reactome overrepresentation analysis results for proteins clustering at each axis end are provided in boxes. Density plots (top and left) illustrate the distribution of human and mouse proteins across both components, while boxplots indicate each component's human vs. mouse protein content per sample. Boxplots show median abundance (center line), interquartile range (IQR, extending from the 1st to the 3rd quartile, box), 1.5 IQR (whiskers), and individual outliers beyond these bounds. Green color indicates murine origin, pink color indicates human origin. Source data are provided as a Source Data file.

## Methods

All research complied with relevant ethical regulations, as detailed in the corresponding sections. A detailed description of patients, materials, and methods is provided in the Supplementary file.

### Patient cohorts and consent

Both cohort studies comply with the Declaration of Helsinki, and all patients provided written informed consent. The MSKCC study was approved by the MSKCC institutional review board (protocol number

**a**

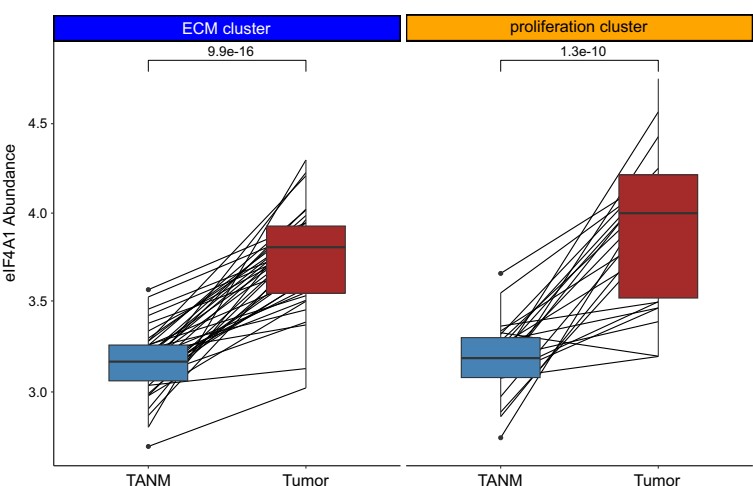

**b**

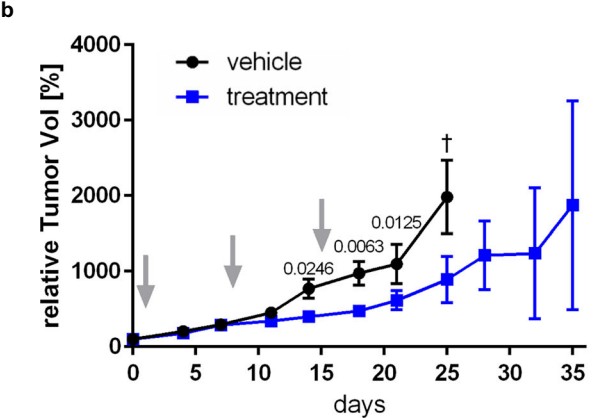

**Fig. 8 | eFT226 treatment. a** Median-normalized abundance difference of EIF4A1 between tumor and TANM samples in the ECM ($n = 48$) and proliferation ($n = 32$) clusters in the human cohort. Unpaired, two-sided $t$-test. Boxplots show median (center line), interquartile range (IQR, extending from the 1st to the 3rd quartile, box), and 1.5 IQR (whiskers). Blue color indicates TANM, red color indicates tumor tissue. **b** Relative changes in PDX tumor volume over treatment time in percent,

$n_{vehicle} = 4$, $n_{treatment} = 4$. Gray arrows indicate treatment time-points. Blue indicates treatment, black indicates vehicle. Unpaired, two-sided $t$-tests, error bars indicate standard error of the mean (SEM). **c** Relative mouse body weight changes over time in percent, $n_{vehicle} = 4$, $n_{treatment} = 4$. Gray arrows indicate treatment time-points. Blue indicates treatment, black indicates vehicle. All unpaired, two-sided $t$-tests, error bars indicate SEM. Source data are provided as a Source Data file.

16-1683 A (3)), and the UKF study was approved by the Institutional Ethics Committee (protocol number 21-1684).

**MSKCC-ICC.** Resected ICC tissues from 80 previously untreated patients were collected at Memorial Sloan Kettering Cancer Center (MSKCC) between 2009 and 2018. Detailed patient and clinical information is available in Supplementary Data 1 and in a histopathological evaluation of the same cohort[18]. Formalin-fixed, paraffin-embedded (FFPE) specimens were prepared and evaluated by the MSKCC department for pathology.

**UKF-ICC.** As a validation cohort, resected ICC tissues from 62 patients were collected at the Medical Center–University of Freiburg (UKF) between 2000 and 2022. Detailed patient information is provided in Supplementary Data 9. FFPE specimens were prepared and evaluated by the Comprehensive Cancer Center Freiburg tumor tissue bank and provided as punch cores of 1.0 mm diameter.

### Sample preparation for LC-MS/MS
MSKCC-ICC tissues were deparaffinized and macrodissected to separate tumor from tumor-adjacent, non-malignant TANM tissue. Proteins were extracted using heat and an acid-labile surfactant. Subsequently, an in-solution digestion protocol was performed with trypsin to generate peptides, essentially as published previously[68].

For the UKF-ICC cohort, a single-pot, solid-phase-enhanced sample preparation (SP3) protocol on an automated liquid handling platform was performed, essentially as published previously[69].

### Patient-derived xenograft proteomics
All animal experiments were approved by the Committee on the Ethics of Animal Experiments of the regional council (Permit Number I-19/02). FFPE tumor sections from nine different patient-derived ICC xenografts (PDX) were provided by Charles River Laboratories and prepared using the S-Trap digestion protocol to generate tryptic peptides.

## LC-MS/MS-based proteomics

For the MSKCC-ICC cohort, 500 ng of peptides were measured on a Bruker TimsTOF Flex mass spectrometer coupled to an Evosep One chromatography system. All measurements were performed using the vendor's 30 samples per day method in data-independent acquisition (DIA) mode. Library prediction and peptide-to-spectrum matching were performed using DIA-NN 1.9.2 with a human proteome database. Semi-specific analyses were conducted with a modified human database containing all possible C- and N-terminal sequence truncations. All expression data were log2-transformed and median-centered.

For the UKF-ICC cohort, 800 ng of peptides were measured on a Thermo Q Exactive Plus mass spectrometer coupled to a Thermo Easy-nLC 1000 chromatography system. All measurements were performed in DIA mode. Library generation and peptide-to-spectrum matching for both canonical and semi-specific analyses were performed as described above using DIA-NN 1.9.2.

PDX samples were measured via the same workflow as the UKF-ICC cohort and analyzed via DIA-NN 1.7 using a combined human-mouse proteome database.

Proteomic data is available at the MassIVE repository (MSV000095336 [https://doi.org/10.25345/C5SN01G2V]).

## Whole-exome sequencing (WES)

DNA from tumor and TANM samples was extracted, libraries prepared and captured (Agilent v6), and subsequently sequenced on the Nova-Seq X Plus platform (PE150). Reads were aligned to GRCh38.p13 with BWA and processed with Sambamba and Picard. Somatic variants were called with MuTect and Strelka and annotated using ANNOVAR. Variants were filtered for coverage and variant allele fraction as described in the Supplementary Methods. WES Data is available via the European Genome-Phenome Archive (EGAD50000001926 [https://ega-archive.org/datasets/EGAD50000001926]).

## Cell culture

Three ICC cell lines (SNU-1079, HuCCT-1, HuH-28) were cultured in an incubator at 37 °C and 5% $CO_2$ in RPMI medium containing 10% fetal bovine serum and 1x penicillin/streptomycin. Cells were treated with different concentrations of eFT226 with or without rapamycin or gemcitabine at fixed concentrations of 200 nM rapamycin for all three cell lines or 1 μM gemcitabine for SNU-1079, and 10 μM gemcitabine for HuCCT-1 and HuH-28. The cell's metabolic activity was measured as a proxy for overall viability using the MTT assay.

## PDX treatment study

All animal experiments were approved by the Committee on the Ethics of Animal Experiments of the regional council (Permit Number G-20/163). NMRI nu/nu mice were implanted with ICC tumors and randomized into experimental groups ($n = 4$ per group) once tumors reached volumes between 75–240 mm³. eFT226 was administered intravenously at a dose of 1 mg/kg once weekly for three weeks. Tumor volumes were measured using a caliper, and animals were sacrificed if tumor volumes exceeded 2000 mm³.

## Data analysis

All statistical analysis was performed in R using RStudio and custom-built scripts. Unsupervised statistical analyses including hierarchical clustering and principal component analysis (PCA) were performed in the MixOmics environment. Differentially expressed proteins were identified using linear models via the Limma package. Enrichment analyses were performed using ClusterProfiler with KEGG, Reactome, and Gene Ontology databases. All classifiers were built with the xgboost package. Survival statistics, including Kaplan-Meier and Cox proportional hazards models, were applied via the survival and survminer packages. In all statistical tests, $p$-values below 0.05 were considered significant, if not indicated otherwise. Scripts are available via GitHub (https://github.com/SchillingLabProteomics/ICC)[70].

## Reporting summary

Further information on research design is available in the Nature Portfolio Reporting Summary linked to this article.

## Data availability

All mass spectrometry-based proteomics datasets used and/or analyzed during this study were uploaded to the MassIVE repository (MSV000095336 [https://doi.org/10.25345/C5SN01G2V]). Genomic raw files were uploaded to the European Genome-Phenome Archive (EGA; EGAD50000001926). Genomic data are subject to restricted access due to their sensitive nature. Access can be requested through our Data Access Committee (https://ega-archive.org/dacs/EGAC50000000048), and requests will be addressed within two weeks. Source data are provided with this paper.

## Code availability

All code is available on GitHub (https://github.com/SchillingLabProteomics/ICC[70] and https://github.com/SchillingLabProteomics/ProteoBoostR)[71].

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

## Acknowledgements

The authors thank Dr. Thien-Ly Dinh for her advice on illustrating proteome regulation in Xenografts. OS acknowledges funding by the Deutsche Forschungsgemeinschaft (DFG, projects 438496892, 546330039, 524803248, 446058856, 507957722, 431336276, 43198400 (SFB 1453 "NephGen"), 423813989 (GRK 2606 "ProtPath"), 322977937 (GRK 2344 "MeInBio"), the ERA PerMed program (BMBF, 01KU1916, 01KU1915A), the ERA TransCan program (BMBF 01KT2201, "PREDICO"; 01KT2333, "ICC-STRAT"), the ERA PerMed project "PerCareGlio" (BMG 2525FSB003), the German Consortium for Translational Cancer Research (project Impro-Rec), the investBW program (project BW1_1198/03 "KASPAR"), the BMBF KMUi program (project 13GW0603E, project ESTHER), the BMBF project "HOLOPROTEOME" (13N17548), nanodiag (03ZU1208AA nanodiag BW), and the MSCA-DN "REMOD-HEALING".

## Author contributions

O.S. and P.B. designed the project; P.B., O.S., M.W., P.H., C.S., and L.T. assembled the cohorts and collected clinico-pathological data; K.B., J.H., K.K., and P.B. performed pathological assessment; K.B., J.R., J.T., and T.W. processed human samples; T.W. processed xenograft samples; T.W. and J.T. performed mass spectrometric measurements; T.W., J.T., M.C.C., F.H., A.T., and N.P. analyzed human data; F.H., T.W., J.S., G.G., and O.S. analyzed xenograft data; N.P., J.T., O.S. analyzed genomics data; T.W., J.T., P.B., and O.S. wrote the manuscript.

## Funding

## Competing interests

O.S., P.B., T.W., K.B., and M.C.C. have filed a patent application on ICC risk-stratifying protein markers. The remaining authors declare no competing interests.
