## [Transparent Peer Review file · Nature Communications]

Proteomic Characterization of Intrahepatic Cholangiocarcinoma Identifies Risk-Stratifying Subgroups and EIF4A1 as a Therapeutic Target

Corresponding Author: Professor Oliver Schilling

Version 0:

Reviewer comments:

Reviewer #1

(Remarks to the Author)
Please refer to the attached file

(Remarks on code availability)
N/A

Reviewer #2

(Remarks to the Author)

The study by Tilman et al. provides a proteomic characterization of intrahepatic cholangiocarcinoma (ICC) in a Western cohort, identifying two subclusters with distinct times-to-recurrence (TTR) and proposing EIF4A1 inhibition as a potential therapeutic strategy for ICC progression. While the study may contribute to the understanding of ICC molecular heterogeneity, it falls short of meeting the high standards expected for publication in Nature Communications, particularly in terms of technical depth, biological novelty, and clinical relevance. Below are the key concerns.

1. The study reports an average of 2,330 identified proteins per sample, with only 1,700 proteins used for downstream analysis. This level of proteomic depth is significantly lower than current standards in the field and may result in the loss of critical protein information, potentially compromising the biological conclusions. Although the authors mention that PDX FFPE samples yielded ~4,600 proteins, this still falls short of the depth achievable with modern proteomic technologies (e.g., TMT or DIA-based approaches). Given the limited depth of identification, the conclusions drawn may not be entirely credible. It is recommended to employ more advanced proteomic methods to improve coverage and ensure robust biological insights.
2. The figures in the manuscript, including main and supplementary figures, appear rudimentary and lack the level of detail and polish. Panels are sparse, and the overall presentation does not meet publication standards.
3. While the study claims to focus on a Western ICC cohort, it fails to provide a meaningful comparison or discussion of how the molecular features of Western ICC differ from those in Asian or other populations.
4. The identification of EIF4A1 as a potential therapeutic target lacks novelty, as the mTOR pathway and its downstream effectors, including EIF4A1, have been extensively studied in various cancers. The study does not provide new mechanistic insights into EIF4A1's role in ICC.
5. The authors utilized PDX models to investigate the interactions between tumor cells and stromal cells in the ICC tumor microenvironment. However, since the stromal cells are derived from mice and the PDX model mice are immunodeficient, this significantly differs from the tumor microenvironment in human patients. Thus, it does not accurately recapitulate the tumor microenvironment in human ICC.
6. The methods section is overly concise and lacks the necessary detail.

(Remarks on code availability)

Reviewer #3

(Remarks to the Author)

(Remarks on code availability)

Reviewer #4

(Remarks to the Author)

In this manuscript, Werner et al. present a DIA proteomic analysis of 80 treatment-naive ICC tumors and adjacent tissues, identifying two prognostic proteomic subtypes, validating their relevance in external datasets, and proposing EIF4A1 as a potential therapeutic target through functional studies in ICC cell lines and patient-derived xenografts.

This study presents an important and valuable dataset, marking the first large-scale proteomic profiling of intrahepatic cholangiocarcinoma in a European ancestry-dominated cohort. The authors' effort in collecting and analyzing these unique samples is commendable, and the data generated has significant potential to advance the understanding of ICC biology.

However, I found the manuscript challenging to read and follow. The central research question remains unclear, and it is not evident how this study addresses a gap that has not already been explored elsewhere. Furthermore, many transitions between sections are abrupt, making the narrative difficult to follow. Strengthening the clarity and flow of the manuscript, along with providing a clearer rationale and interpretation of the findings, would greatly enhance its impact.

The section on TANMs feels abrupt. The authors note that the Dong et al. (2022) FU-ICC paper excluded TANMs, but the findings in this study seem subtle. It appears to support the idea that omitting TANMs does not make much difference, rather than offering new insights.

In section 3.5, the description of how the four proteins were selected is surprisingly vague. The criteria include "large quantitative differences"—but how large? "Low adjusted p-values"—how low? What statistical method was used to compare their abundances between clusters? Were confounding variables adjusted? What thresholds were applied? Additionally, the rationale for using these four proteins to predict the ECM/turnover clusters requires clarification.

The proteome-wide survival analysis also appears suddenly. The authors mention a "second analysis," but it is unclear why this additional analysis was necessary, given that patient clustering already demonstrated significant clinical relevance. This section lacks a clear conclusion, offering only a list of individual significant proteins.

I found the semi-specific peptide analysis compelling. It suggests altered protease activity in tumors of the ECM cluster, with a strong signal. However, this paragraph also appears abruptly and would benefit from better integration with the prior section. Several important questions remain:

1. What are the proteins from which the tumor-only truncated peptides are derived?
2. Are they primarily ECM-related proteins? If so, is the observation simply that ECM tumors have more ECM proteins vulnerable to protease activity, rather than reflecting perturbed protease activity itself?
3. Which proteases are responsible? Are their abundances different between groups?
4. Is the abundance or proportion of truncated peptides associated with clinico-pathological variables?
5. Is there a similar pattern in the FU-ICC cohort? Could ancestry differences play a role?

The legend of Figure 6C describes the PLS-DA as based on truncated "peptide expression," but lines 387–388 refer to "truncated proteins," which is an improper interpretation. Also, given the expected sparsity of truncated peptide identifications across samples, it is unclear how informative the PLS-DA really is beyond demonstrating that "they are somewhat different."

The PDX analysis and the MSKCC cohort proteomics seem disjointed. Are the nine PDX samples described in section 3.9 derived from the same patients as the eight PDX used for the drug analysis in section 3.10? How are they related to the MSKCC-ICC cohort? I am not convinced of the added value of the PDX proteomics analysis in its current form.

In the conclusion, the authors highlight this as the first large-scale ICC proteomic study with European ancestry-dominated patients. However, the core power of this comparison is not clearly demonstrated. While differences between tumor and NAT protein abundances are shown, this is not surprising.

The FU-ICC cohort is used as an external validation dataset, but it is a Chinese-dominated cohort. Do genomic profiles differ between ICC patients of Chinese versus European ancestry? If so, how might this influence the proteome? The statement that "genomic analysis [was] not feasible" is not convincing. More comparative analysis could be performed to evaluate differences in tumor/normal proteomes and survival-associated proteins (section 3.7) between the two cohorts.

Additionally, the number of proteins identified per patient appears somewhat low compared to the FU-ICC cohort, which raises concerns about the depth and completeness of the proteome coverage in this study.

Minor Concerns and Suggestions:

Figures are not cited correctly. For example, line 250 cites Figure S2, but it should be S3. Figures are also cited out of order, such as Figure S1C being cited after Figure S2, which is confusing.

The proteomics data analysis requires clearer explanation. It is not clear how the spectral library was generated. The manuscript mentions a pooled sample was used, presumably for DDA, but no details are given about the method or data analysis.

Why was a relatively high FDR (10%) used during spectral library generation? Please clarify the method and justification.

Statistical results are often incomplete. P-values and/or effect sizes/HR are missing in many places, including lines 215 and 222.

Statistical analysis approaches are not explained in sufficient detail. For survival analysis (section 3.7), the authors mention using the survival and survminer packages, but what confounding variables were adjusted for? Section 3.7 mentions adjusting for adjuvant radiotherapy, but were adjustments made for ECM/turnover clusters in section 3.5? Why were other variables, such as stage, sex, age, and virus status, not adjusted? These factors were mentioned in the introduction as relevant risk factors. The FDR correction method for proteome-wide survival analysis is also not described.

For the differential abundance analysis of tumor versus normal, were data scaled before fitting the linear model? Did the model include random intercepts for each patient? Were confounding variables considered? How was FDR controlled?

In the PCA plot (Figure 1F), PC1 explains an unusually high percentage of variance. Were data scaled before running PCA? If not, the difference may be driven by high-abundance proteins.

Line 246: "PCA of proteome data" is vague. "PCA of protein abundance" may be clearer.

Line 248: The interaction between PCA results and gender, age, or stage is not shown in Figure 1F.

Line 249: The authors state that proteomic clustering was confirmed by MC simulation. It is unclear whether they (1) used hierarchical clustering and found consistency with M3C, or (2) used M3C to determine two clusters, then applied hierarchical clustering. If (1), the consistency between methods should be shown; if (2), the wording should be clarified.

Line 264: The authors mention gender. Do they mean gender or sex? If referring to biological characteristics, "sex" is more appropriate.

Line 265: Missing p-value cutoff and statistical test used.

Line 267: Missing p-value.

Line 268: Assumed pathway enrichment results, but the description lacks clarity.

Line 272: I believe the authors meant to refer to the "cluster" rather than "cluster association." It is unclear what association is being referenced, and why such an association would be predictive of an outcome. Additionally, it would be helpful to clarify whether confounders were controlled for in this analysis. A significant proportion of patients in the turnover group received adjuvant radiotherapy—were these patients presenting with more aggressive disease, potentially leading both to the decision for adjuvant radiotherapy and to worse survival outcomes?

Line 317: What is the hazard ratio?

(Remarks on code availability)

I appreciate the authors' effort to support open science and reproducibility by providing code in the associated GitHub repository. However, the repository appears to only include scripts and R Markdown files for generating the figures, and it seems that some supplementary figures are not covered. Additionally, it appears that all commits were pushed within a two-day window, which means that the authors may have either selectively uploaded certain scripts to the repository specifically for publication or did not employ version control throughout their analysis. I encourage the authors to provide a more comprehensive and transparent workflow, ideally with version-controlled scripts that cover the entire analysis process, which would significantly enhance reproducibility.

Version 1:

Reviewer comments:

Reviewer #1

(Remarks to the Author)

The authors have provided comprehensive responses to the concerns raised by the reviewers, and the authors are to be commended for conducting additional mass spectrometry data acquisition runs using a newer model mass spectrometer to increase the proteomic depth of their MSKCC-ICC cohort. The quality of the manuscript has improved significantly. On clarifying question and one suggestion:

1. Panel A of the figure on pg. 1 of the response to reviewers refers to "Pearson correlation R2." It is unclear what statistical parameter the authors are referring to here given that the Pearson correlation is represented by (r), not R2. Similarly, on pg. 2, of their response, the authors mention, "Pearson correlation of R2 = 0.81, whereas "r" is mentioned in the correlation plots below the text.
2. Consider adding the figures on pg. 5 of the response to the concerns raised by the reviewers to the Supplementary Data. The figures provide informative data re: the association between tumor cellularity vs. PFS and OS.

(Remarks on code availability)

Reviewer #2

(Remarks to the Author)

I have carefully reviewed the authors' revised manuscript and their point-by-point response. While I appreciate the authors' additional experiments and revisions, several important issues remain insufficiently addressed, as detailed below.

1. The authors reanalyzed their peptide samples using the TimsTOF, which increased the number of identified proteins from ~2,000 to over 7,000. This represents a substantial improvement in proteome coverage. However, the subsequent analyses and biological conclusions remain essentially identical to those presented in the previous version. Given this significant increase in data depth, one would expect additional findings or refined biological insights into ICC. Otherwise, this raises the question of whether a much lower proteome coverage would have been sufficient to address the same biological questions. The current revision does not adequately explain why such a dramatic increase in protein identification did not lead to new or strengthened conclusions.
2. The authors' point-by-point responses are largely textual and do not provide corresponding figures, figure legends, or manuscript excerpts to demonstrate where and how each issue was addressed. This makes it difficult to evaluate whether and to what extent the reviewers' comments were incorporated into the revised manuscript. For transparency and clarity, the authors should clearly indicate the exact locations of revisions in the text and include updated figures or supplementary materials where relevant.
3. The authors claim that the role of EIF4A1 in ICC is being reported for the first time. However, recent literature has already described this association — for example: Mi W, Cigliano A, Galleri G, et al. Targeting EIF4A1 is effective against human intrahepatic cholangiocarcinoma. *JHEP Rep.* 2025;7(7):101416. Furthermore, the study does not provide new mechanistic insights into EIF4A1's role in ICC, which weakens the novelty of this finding.
4. The authors state that "the PDX models have particular value in corroborating the origin (tumor vs. stromal cells) of the various matrix components." However, it remains unclear what specific new conclusions were derived from these experiments. The description is rather general, and neither the main text nor the figures provide sufficient detail to support this claim.
5. Each figure title is currently written as a methodological phrase, rather than summarizing the main finding or conclusion of the figure. This makes it difficult for readers to follow the logical flow of the study or grasp the key results.

(Remarks on code availability)

Reviewer #3

(Remarks to the Author)

(Remarks on code availability)

Reviewer #4

(Remarks to the Author)

I really appreciate the authors taking the time to perform new experiments and analyses. I am very glad to see that the manuscript has significantly improved, especially in terms of data quality and interpretability. I now like the semi-specific peptide analysis even better. With the new figures, it is now clearer that certain proteins undergo increased proteolytic digestion, particularly in the proliferation-associated cluster. This is a fascinating observation. It raises

many interesting follow-up questions that, while outside the scope of this current paper, could inspire future work. One suggestion I would offer is to explore whether this ECM vs. proliferation tumor subtype classification can also be observed at the transcriptomic level. Since RNA-seq data is available, it would be informative to ask whether ECM- or proliferation-related genes show elevated expression in their respective clusters. Including a figure or two illustrating this could make the picture of the study even more complete.

The only major issue I would still like the authors to address is the lack of detail in the WXS methods section. Please provide specifics such as the capture kit used, the reference genome version (e.g., GRCh38 p14?), the alignment tool, the somatic mutation caller, filtering criteria, and annotation algorithm. The phrase "standard criteria" is way too vague.

Additionally, when discussing mutation burden (e.g., "~30 mutations per tumor"), the authors should report TMB as the number of mutations per Mbp of the capture region. The number of somatic mutations is dependent on the capture design.

(Remarks on code availability)

Seems like only analysis code for figure generation was included.

“Proteomic Characterization of Intrahepatic Cholangiocarcinoma Identifies Risk-Stratifying Subgroups, Proteins Associated with Time-To-Recurrence, and mTOR Effector Molecule EIF4A1 as a Druggable Therapeutic Target”

Response to reviewers - Preface for all reviewers

We thank all reviewers for their extensive comments and suggestions to improve our manuscript. As outlined in the point-by-point response, we are addressing all concerns. One issue raised by all reviewers has been the proteomic depth of the MSKCC-ICC cohort. We agree with this concern and re-measured stored peptides from our MSKCC-ICC cohort on a state-of-the-art Bruker TimsTOF Flex mass spectrometer coupled to an Evosep One chromatography system.

The new proteomic data yielded an average of 7020 identified proteins per sample, more than 5500 of which were identified consistently at less than 20% missing values across the cohort. Virtually all

Correlation QExactive - TimsTOF measurements: A) Pearson Correlation coefficients R^2 for all samples in the cohort. B) Overlap of identified proteins. C) Overlap of identified proteins with less than 20% missing values. D&E) Principal Component Analyses (PCA) of the MSKCC-ICC cohort based on the new TimsTOF measurements. New clusters in D, old QExactive clusters in E.

proteins that had been detected in the old measurements (using a Thermo QExactive mass spectrometer) could be identified again.

Looking at quantitation, we found a mean Pearson correlation of $R^2 = 0.81$ when comparing old and new data for proteins that were identified in both measurements. Therefore, we conclude that the underlying proteomic information remained intact, even as overall protein identifications have tripled. Two proteomic clusters stratifying ICC tumors along biological themes (extracellular matrix versus

Correlation QExactive - TimsTOF measurements: Correlation Plots for all MSKCC-ICC cohort samples including all proteins identified in both measurements. Pearson Correlation incl. R^2 and p-value given.

mRNA and protein turnover) and progression-free survival were the core findings in our original data. And indeed, hierarchical clustering of the new data again resulted in two proteomic clusters displaying significant differences in the time-to-recurrence (TTR). The old and new clusters are consistent in their biological patterns, with the much-improved proteomic depth redefining the classification of a few select samples. One cluster is defined by a high content of extracellular matrix proteins and longer TTR (ECM-cluster), and the other showing higher expression of proteins involved in DNA replication and the synthesis and processing of mRNA and proteins. Because this marked upregulation of DNA replication proteins had not been visible in our old data, we decided to rename the second cluster as “proliferation cluster”.

Other results from the manuscript's first version remained comparable as well and benefited from the new proteomic depth. For instance, similar proteomic patterns were found to directly relate to progression-free survival in the Cox proportional hazards analysis, with more than half of the originally identified proteins again being significant hits (25 out of 39). Only the semi-specific analysis shifted in focus, as we now observe significantly more semi-specific peptides in TANM tissue, compared to tumors. This might be due to the increased proteomic depth of the new measurements, leading to a notable increase in the number of identified semi-specific peptides (identified semi-specific peptides old = 868 versus new = 18087). Nevertheless, although the quantity of semi-specific peptides does not differ between the clusters, they do still differ in quality.

Remarkably, when we transferred the new clusters to the Fudan ICC (FU-ICC) cohort via a refined classifier approach, we again detected a significant difference in patient survival between the assigned ECM- and proliferation clusters. Encouraged by these results, we procured another, wholly independent ICC cohort from our own institution (UKF-ICC) and stratified it into both clusters using the same classifier workflow. Also here, the ECM-cluster was significantly linked to longer overall survival.

Point-by-Point Response

Reviewer 1

Intrahepatic cholangiocarcinomas (ICCs) are rare cancers that originate in bile ducts and ductules within the liver. These cancers have non-specific symptoms and there is a lack of reliable, noninvasive prognostic biomarkers. Using a mass spectrometry-based proteomic approach, the authors analyzed 80 treatment-naïve ICC tumors, 77 adjacent non-malignant tissues, and 9 PDXs. Two proteomic sub-clusters, extracellular matrix (ECM) and mRNA/protein turnover, exhibited distinct times-to-recurrence (TTR). These prognostic proteomic clusters were validated using a separate dataset, and EIF4A1 inhibition was identified as a potential strategy to mitigate ICC progression.

The authors generated a significant amount of data for this study; however, there are concerns regarding the data quality and its subsequent impact on the biological interpretation of the data. Several major and minor concerns need to be addressed before this reviewer would consider recommending the manuscript for publication.

Answer: We sincerely thank the reviewer for their evaluation of our work and for highlighting both the strengths of the study and the concerns regarding data quality and interpretation. We have addressed each point in detail below and believe that the clarifications and additional analyses provided will help to resolve these concerns.

Major

1. The font sizes in several of the figures (Fig. S1A & B, 5A & E-F, S3, S5, S7) are too small, which consequently precludes an accurate assessment of the presented data. If the font sizes in the figures cannot be reasonably enlarged, consider reducing the content displayed in each figure to only include the most salient aspects.

Answer: Thank you for this comment. We agree and apologize for the original figures being hard to read. In the scope of our new analysis, we have updated all figures in the manuscript to reach publication standards.

2. How do the eFT226 and Gemcitabine concentrations used to treat the ICC cell lines relate to IC₅₀ values?

Answer: We thank the reviewer for this important point. The concentrations of eFT226 and Gemcitabine used in our ICC cell line experiments were chosen to cover a range around the IC₅₀ values determined from dose-response curves (fitted using a four-parameter logistic model in GraphPad Prism). For eFT226, the IC₅₀ values were 11.18 nM (HuH-28), 8.53 nM (HuCCT-1), and 4.68 nM (SNU-1079). Notably, HuH-28 showed only partial sensitivity to eFT226, with maximal inhibition reducing viability to ~60%, and is thus regarded as partially resistant. In this case, the classical IC₅₀ corresponds to the midpoint between the fitted Top (100%) and Bottom (~60%), while the concentration required to reduce viability to 50% of control exceeds the highest dose tested. For Gemcitabine, the experimental doses (1 μM and 10 μM) were chosen to ensure measurable effects of around 50% reduction of cell viability across both sensitive (HuCCT-1 and SNU-1079) cell lines. As the HuH-28 cell line was considered resistant, the higher dose of 10 μM was applied, acknowledging that in resistant lines the 50% viability threshold may not have been reached within this range. Calculated Gemcitabine IC₅₀ values as mid-points of half-maximal inhibition between the fitted top and bottom plateaus were 1.052 μM (HuCCT-1) and 0.005 μM (SNU-1079).

3. It does not appear that the authors determined the tumor cellularity of the ICC patient tissues. This is a common drawback of the –omic analysis of bulk tissues.

Answer: Thank you for this insightful comment. It is true that the high ECM content in the ECM-cluster could suggest simply a lower number of tumor cells relative to the proliferation cluster. We have consulted a trained pathologist, who has used an imaging classification model (QPath) to identify and count different cell types within scans of the used tumor slides. We found no significant

difference in the relative tumor cell content between both clusters (see manuscript 3.4 and Table 1). Neither was the tumor cellularity associated with the time-to-recurrence or patient overall survival.

Cox Proportional Hazards Model: Impact of Tumor Cell Content [%] on left) progression-free survival and right) overall survival.

4. A log fold-change cut-off of 0.5 was used for the data presented in section 3.7 regarding the prognostic ICC proteins related to progression-free survival. This is a considerably low cut-off (fold change of ~1.4) to determine whether a protein is “significantly enriched”.

Answer: Thank you for pointing this out. Although we believe that even small log₂-fold changes do provide valuable information if highly consistent across the cohort (as demonstrated by low adjusted p-values), we understand that a cutoff at 0.5 might be too lenient to warrant follow-up investigations. We have thus increased the cutoff to log₂ 0.8 (fold change of 1.75). Additionally, your comment points toward one issue of Cox proportional hazards model (CPHM) we decided to address directly in the main text: CPHM only regards how well protein abundances correlate with progression-free survival but does not consider absolute abundance changes. Hence, even subtle changes in protein abundance can count as significant, as long as they sufficiently align with survival data. For this reason, CPHM results should be viewed in conjunction with abundance-based analyses such as differential expression analysis or a simple look at the coefficient of variation of protein abundances across the cohort. We have now also added this data to our manuscript.

5. The interpretation of semi-tryptic peptides as evidence of “endogenous proteolytic processing events” is dubious. No evidence is provided to enable the differentiation of biologically relevant “endogenous proteolytic processing events” from technical/analytical artefacts. The experimental design for this study was not set up to enable the analysis of endogenous proteolytic processing events.

Answer: We agree with the reviewer that our original phrasing has been ambiguous as semi-specific peptides are not linked to proteolytic processes by default. Still, proteolysis is widely known to shape tumor progression, especially by remodeling the extracellular matrix¹ and proteolysis is the major contributor to semi-specific peptides^{2,3}. Similar computational strategies have been applied in recent proteomic studies to infer protease activity and tumor-associated proteolysis^{2,4,5}. Our reanalysis shows biological enrichment consistent with tumor biology, suggesting at least part of the signal reflects endogenous proteolytic processes.

6. Fig. 2A: There are significant concerns regarding the validity of the number of differentially regulated proteins; $1,368/1,700 = 80\%$! If the data were normally distributed, only 32%, 4.5%, or 0.3% of the proteins would have a relative abundance exceeding the mean ± 1 sd, 2 sd, or 3 sd, respectively. It is possible that the protein profiles of TANM vs. tumor are markedly different; however, one would not expect 80% of the proteins to be differentially abundant.

Answer: We thank the reviewer for this attentive observation. After remeasurement of the whole cohort resulting in higher proteome coverage, we found 4865 out of 5587 proteins (87%) to be differentially abundant between TANM and tumor tissue. We agree that the proportion of differentially abundant proteins appears high at first sight. However, as confirmed by the responsible pathologist, the tumor and TANM tissues represent markedly different cellular and morphological compositions, despite being located adjacently on the same slide. This pronounced contrast is further reflected in the PCA (Fig. 1E), which demonstrates clear separation of the two groups. Therefore, the large fraction of proteins identified as differentially abundant is consistent with the underlying biological differences between the tissues, rather than an artifact of the data analysis. Moreover, in our UKF-ICC cohort, we see similar results, with 2581 out of 3064 proteins (84%) to be differentially abundant between TANM and tumor tissue.

7. It is unclear why the authors identified $>2x$ more human proteins when analyzing the PDX tissue (3,168) vs. the patient tissue (1,368) if the sample processing and analytical methods were indeed similar. The patient tissues and the PDX tissues were FFPE specimens.

Answer: We thank you for pointing out this contrast in the number of identifications. Our re-measurement of the MSKCC-ICC cohort addresses this issue and has boosted overall identifications and quantifications. For context, the original cohort measurement had been performed using self-packed LC-columns (length approx. 35cm), whereas PDX samples were measured using a 2m micro-PAC column (ThermoFisher), which strongly improved separation efficiency and downstream protein identifications.

Minor:

1. Figure 1E is overly-convoluted/complicated and no key is provided explaining the symbols in the middle of the PCA plots. Is it important to have a unique symbol representing each distinct sample?

Answer: We agree and have removed the figure, since batch correction has not been necessary for the new data. For context, the old data was acquired in the course of more than three weeks in more than 20 batches. New data using a Bruker TimsTOF Flex coupled to an Evosep One system could be measured within less than 5 days in only 3 batches.

2. Fig. S2: Add a legend for the red-blue color gradients/scales.

Answer: Thank you for this observation! We have added a legend detailing the unit and scale of the color gradient to all KEGG enrichment plots.

3. In the legend for Fig. 8, “dotted lines” are referred to as indicating treatment time-points, but they do not appear in the figure. Moreover “grey arrows” are also mentioned as indicating treatment time points.

Answer: We have revised the figure for clarity. Now only grey arrows indicate treatment time points.

4. Titles should be included with each Supplementary Table.

Answer: Thank you. We have taken up this suggestion for the new supplementary tables.

Reviewer 2

The study by Tilman et al. provides a proteomic characterization of intrahepatic cholangiocarcinoma (ICC) in a Western cohort, identifying two subclusters with distinct times-to-recurrence (TTR) and proposing EIF4A1 inhibition as a potential therapeutic strategy for ICC progression. While the study may contribute to the understanding of ICC molecular heterogeneity, it falls short of meeting the high standards expected for publication in Nature Communications, particularly in terms of technical depth, biological novelty, and clinical relevance. Below are the key concerns.

Answer: We thank the reviewer for their thoughtful assessment of our work and for highlighting the shortcomings of the previous version. We greatly appreciate the constructive feedback and have carefully revised the manuscript to address all points of concern. We hope that the current version more convincingly demonstrates the technical depth, biological novelty, and clinical relevance of our study.

1. The study reports an average of 2,330 identified proteins per sample, with only 1,700 proteins used for downstream analysis. This level of proteomic depth is significantly lower than current standards in the field and may result in the loss of critical protein information, potentially compromising the biological conclusions. Although the authors mention that PDX FFPE samples yielded ~4,600 proteins, this still falls short of the depth achievable with modern proteomic technologies (e.g., TMT or DIA-based approaches). Given the limited depth of identification, the conclusions drawn may not be entirely credible. It is recommended to employ more advanced proteomic methods to improve coverage and ensure robust biological insights.

Answer: We do agree with your observation and have thus decided to re-measure the entire cohort as detailed above. We now identified on average 7,020 proteins per sample and proceeded with 5,500 proteins for downstream analysis. We are confident that the new proteome coverage is on-par with current standards and provides a lot deeper biological insight.

2. The figures in the manuscript, including main and supplementary figures, appear rudimentary and lack the level of detail and polish. Panels are sparse, and the overall presentation does not meet publication standards.

Answer: We apologize for the lower figure quality in our original version. We have now updated all figures to be ready for publication.

3. While the study claims to focus on a Western ICC cohort, it fails to provide a meaningful comparison or discussion of how the molecular features of Western ICC differ from those in Asian or other populations.

Answer: We appreciate this important point and acknowledge that significant population differences exist in ICC pathogenesis and molecular features. However, our key finding—the presence of a low-risk ECM-enriched subtype and a high-risk proliferation subtype—was robustly validated in the independent Fudan cohort representing an Asian population, demonstrating the generalizability of these proteomic subtypes across populations. ICC pathogenesis indeed differs between populations due to varying etiological factors. Chronic viral hepatitis and liver fluke infections are reported to be more prevalent in Asian regions, whereas metabolic-associated liver diseases and other factors predominate in Western countries⁶. These different etiologies likely result in diverging genomic profiles and histopathological features⁷⁻¹¹. For example, studies indicate that Chinese populations show more genetic aberrations per patient and more mutations in DNA repair pathway-associated genes, while American patients exhibit higher frequencies of CDKN2A/B and IDH1/2 genetic aberrations¹². Additionally, Dong et al. reported higher KRAS mutation frequencies but lower IDH1, ARID1A, and TERT mutation frequencies in their Fudan ICC cohort compared to Western cohorts¹³. Despite these population-specific differences, our proteomic classifier successfully identified the same two major subtypes in both Western and Asian populations. This indicates that the fundamental proteomic organization of ICC into ECM-enriched and proliferation-driven subtypes represents a common biological framework.

4. The identification of EIF4A1 as a potential therapeutic target lacks novelty, as the mTOR pathway and its downstream effectors, including EIF4A1, have been extensively studied in various cancers. The study does not provide new mechanistic insights into EIF4A1's role in ICC.

Answer: We agree that the mTOR pathway has been investigated extensively in multiple cancer entities, highlighting its importance as a driver of cancer progression. Various treatments targeting this pathway are available. Yet, EIF4A1 is a translation initiation factor. While mTOR signalling is a major switch for protein biosynthesis, the actual translation process is mechanistically distinct from the signalling axis. Moreover, EIF4A1 inhibitors represent a distinct mode-of-action (blocking translation rather than signalling), and are gaining traction as a new therapeutic target in different cancers. For this reason, we deem it relevant that mTOR signaling in general, and EIF4A1 in particular, also appear to play a role in ICC tumors. EIF4A1 inhibition has so far not been investigated in bile duct malignancies, making our study the first instance. Although a retrospective proteomic study of patient tissues can hardly provide mechanistic insights into EIF4A1 in ICC, we believe there is value in putting EIF4A1 inhibition on the map as a so far unconsidered treatment option for this cancer.

5. The authors utilized PDX models to investigate the interactions between tumor cells and stromal cells in the ICC tumor microenvironment. However, since the stromal cells are derived from mice and the PDX model mice are immunodeficient, this significantly differs from the tumor microenvironment in human patients. Thus, it does not accurately recapitulate the tumor microenvironment in human ICC.

Answer: We agree that patient-derived xenografts (PDX) can only serve as a proxy for tumor progression and thus come with notable limitations. Nevertheless, PDX are well-established models not just for drug testing but for controlled molecular studies involving patient tissues¹⁴⁻¹⁶. In fact, the mice still carry functional innate immune and tissue responses to implanted tumors^{17,18}. In the context of the present study, the PDX models have particular value in corroborating the origin (tumor vs. stromal cells) of the various matrix components.

6. The methods section is overly concise and lacks the necessary detail.

Answer: Thank you for your comment. The full materials and methods section can be found in the supplements.

Reviewer 3

Answer: Thank you for your help!

Reviewer 4

In this manuscript, Werner et al. present a DIA proteomic analysis of 80 treatment-naive ICC tumors and adjacent tissues, identifying two prognostic proteomic subtypes, validating their relevance in external datasets, and proposing EIF4A1 as a potential therapeutic target through functional studies in ICC cell lines and patient-derived xenografts.

This study presents an important and valuable dataset, marking the first large-scale proteomic profiling of intrahepatic cholangiocarcinoma in a European ancestry-dominated cohort. The authors' effort in collecting and analyzing these unique samples is commendable, and the data generated has significant potential to advance the understanding of ICC biology.

However, I found the manuscript challenging to read and follow. The central research question remains unclear, and it is not evident how this study addresses a gap that has not already been explored elsewhere. Furthermore, many transitions between sections are abrupt, making the narrative difficult to follow. Strengthening the clarity and flow of the manuscript, along with providing a clearer rationale and interpretation of the findings, would greatly enhance its impact.

Answer: Thank you for this encouraging assessment and for your helpful critique! We have revised the manuscript to improve its structure and flow. We hope that these changes make the rationale more evident and the manuscript easier to read.

The section on TANMs feels abrupt. The authors note that the Dong et al. (2022) FU-ICC paper excluded TANMs, but the findings in this study seem subtle. It appears to support the idea that omitting TANMs does not make much difference, rather than offering new insights.

Answer: When re-reading our manuscript we came to agree with your observation and have thus removed the section on TANM tissues. Still, for the sake of completeness, we want to mention that

also in the new data TANM clusters remained disconnected from both tumor proteomic clusters and from clinical annotations, highlighting their status as tissues with an apparently rather independent proteome biology.

In section 3.5, the description of how the four proteins were selected is surprisingly vague. The criteria include "large quantitative differences"—but how large? "Low adjusted p-values"—how low? What statistical method was used to compare their abundances between clusters? Were confounding variables adjusted? What thresholds were applied? Additionally, the rationale for using these four proteins to predict the ECM/turnover clusters requires clarification.

Answer: We thank you for this comment. Following your remarks we have reworked the section on how the classifier proteins were selected. Now, they are stringently chosen to include two proteins with the highest log2-fold change and the lowest adjusted p-value representative for each cluster. Additionally we demonstrate that not only few selected proteins can serve as classifiers but that many combinations of the 50 most cluster-representative proteins are suitable to build good classification models. We admit that the decision to test four-protein classifiers was based less on mathematical modeling and more on our clinical background. Working in an Institute for Pathology, we aimed for a protein number sufficient for reproducible stratification of new samples, yet still practical for testing not only by mass spectrometry but also via antibody-based methods.

The proteome-wide survival analysis also appears suddenly. The authors mention a "second analysis," but it is unclear why this additional analysis was necessary, given that patient clustering already demonstrated significant clinical relevance. This section lacks a clear conclusion, offering only a list of individual significant proteins.

Answer: Also here, we are grateful for the constructive feedback! It is true that the original version of the text needed to demonstrate more clearly the additional value of CPHM versus the clustering analysis. We have added that information to the main text. In brief, while the clustering analysis provides an unsupervised description of major biological motifs within the cohort, the CPHM specifically searches for proteins whose expression relates to the time-to-recurrence. Therefore, among the multitude of proteins identified in the clustering analysis, the CPHM serves to highlight biological themes and, most importantly, individual proteins that correlate best with survival.

I found the semi-specific peptide analysis compelling. It suggests altered protease activity in tumors of the ECM cluster, with a strong signal. However, this paragraph also appears abruptly and would benefit from better integration with the prior section. Several important questions remain:

Answer: Thank you for this encouraging feedback. As mentioned above, the detection of semi-specific peptides increased more than 20-fold in the newly measured dataset, thus changing the observable landscape of the "semi-specific peptidome" considerably. We have addressed your questions in light of these new results.

1. What are the proteins from which the tumor-only truncated peptides are derived?

Answer: We repeated the differential expression analysis outlined in with only semi-specific peptides to address your question. A subsequent gene set enrichment analysis using the GeneOntology database highlighted biological processes enriched in either tissue. As visible in the figure below,

most semi-specific peptides detected in tumor tissues originated from proteins involved in proliferation processes including extracellular processes like cell adhesion and supramolecular fiber organization, as well as RNA and DNA processing. In contrast, semi-specific peptides in TANM tissue derived mostly from metabolic processes. Therefore, detected semi-specific peptides at large mostly reflect each tissue's biological fingerprint.

Gene Set Enrichment of semi-specific peptides: left TANM, right tumor.

2. Are they primarily ECM-related proteins? If so, is the observation simply that ECM tumors have more ECM proteins vulnerable to protease activity, rather than reflecting perturbed protease activity itself?

Answer: Considering the new data, we do not observe significant differences in the amount of semi-specific peptides relative to all detected peptides between the clusters (Fig 6E). Overall, the abundance of semi-specific peptides correlates well with the abundance of their parent protein, suggesting proteolytic decay rather than targeted regulation. Nevertheless, a subset of semi-specific products does not follow this correlation and may indicate specific proteolytic processing, such as observed for Aldo-keto reductase family 1 member C4 (AKR1C4).

3. Which proteases are responsible? Are their abundances different between groups?

Answer: Using the MEROPS protease database as reference (<https://www.ebi.ac.uk/merops/index.shtml>), we found 280 proteases (or proteins with putative proteolytic function) within our dataset, many of which have overlapping and poorly distinguishable specificities. At the same time, the semi-specific data represents a mixture of convoluted cleavage events. We deemed it beyond the scope of the present study to bioinformatically deconvolute the “web” of proteolytic events.

We reanalyzed the data shown in the section “Tumor and TANM Tissues Show Highly Divergent Proteome Profiles” to specifically look for proteases that are differentially expressed between tumor and TANM tissues. To this end, we used the MEROPS protease database. We considered proteases to be significantly enriched in either tissue, if they presented an absolute log₂-fold change greater than 0.5 and an FDR-adjusted p-value below 0.05. Using [Reactome.org](https://reactome.org) overrepresentation analysis, we find that most proteases enriched in tumors (88 in total) relate to ECM degradation, neutrophil degranulation, caspases, and innate immune responses. Conversely, proteases that are increased in TANM tissues (in total 66) are mostly involved in metabolic functions, as would be expected in liver tissue.

Volcano Plot Tumor vs. TANM: limma-based differential expression analysis as in main figure 2A. The top 10 proteases with the highest log₂-fold changes (identified via the MEROPS protease database) are named. Blue indicates higher abundance in TANM tissue, red indicates higher abundance in tumor tissue ($p < 0.01$).

We repeated this analysis to also look for differentially expressed proteases between both clusters. Here, we found fewer proteases to show a significant enrichment profile. Again relying on [Reactome.org](https://reactome.org), we found many detected proteases in the ECM-cluster (31 in total) to be involved in the complement cascade. The proliferation-cluster showed a comparative sparsity of proteases, with only 6 being enriched here in absence of a clear biological pattern.

Volcano Plot ECM- vs. proliferation-cluster: limma-based differential expression analysis as in main figure 3C. The top 10 proteases with the highest, significant log₂-fold changes above 0.5 (identified via the MEROPS protease database) are named. Blue indicates higher abundance in the ECM cluster, orange indicates higher abundance in the proliferation cluster ($p < 0.05$).

We have added both plots in Supplementary Figure S11.

4. Is the abundance or proportion of truncated peptides associated with clinico-pathological variables?

Answer: We thank the reviewer for raising this very interesting point. We tested the proportion of truncated peptides against the clinico-pathological variables but did not find any statistically significant associations.

Distribution and correlation of clinicopathological parameters with the proportion of semi-specific peptides. Blue indicates TANM tissue, red indicates tumor tissue.

5. Is there a similar pattern in the FU-ICC cohort? Could ancestry differences play a role?

Answer: Unfortunately, raw mass spectrometry data from the FU-ICC cohort is not publicly available for re-analysis. However, to address this point, we have considered the semi-specific peptides in our additional UKF-ICC cohort. The overall number of semi-specific peptides was lower than in the MSKCC dataset, reflecting differences in measurement setup. Nevertheless, consistent with our

main findings, the proportion of semi-specific peptides was higher in TANM compared to tumor samples in both clusters. We also observed substantial global differences in semi-specific peptide abundances between TANM and tumor tissue, whereas differences between clusters were minimal (see panel C&E), possibly due to lower purity of FFPE punch samples and/or higher sparsity in the semi-specific dataset. Interestingly, RTN4 emerged as the only protein with a peptide not correlating with the abundance of its parent protein - consistent with its behaviour in the MSKCC cohort.

Semi-tryptic analysis of the UKF-ICC cohort: A) Count of fully tryptic (specific) and semi-specific detected peptides. B) Boxplot indicating the ratio of semi-specific peptides among all detected peptides for tumor and TANM samples in the ECM and proliferation cluster. Unpaired, two-sided t-test. Boxplots show median (center line), interquartile range (IQR, extending from the 1st to the 3rd quartile, box), and 1.5 IQR (whiskers). C) Partial Least Squares-Discriminant Analysis (PLS-DA) of semi-specific peptide expression in both clusters and matching TANM tissues. Ovals indicate 95% confidence intervals. D) Correlation of logFC values between protein abundance and semi-tryptic peptide abundance in tumor vs. TANM tissue. E) Correlation of logFC values between protein abundance and semi-tryptic peptide abundance across ECM vs. proliferation clusters.

The legend of Figure 6C describes the PLS-DA as based on truncated "peptide expression," but lines 387–388 refer to "truncated proteins," which is an improper interpretation. Also, given the expected sparsity of truncated peptide identifications across samples, it is unclear how informative the PLS-DA really is beyond demonstrating that "they are somewhat different."

Answer: We fully agree that our original wording was imprecise, the PLS-DA in Figure 6 C is based on semi-specific peptide intensities, not truncated proteins. We have revised the text to clarify this. We also acknowledge that the amount of sparsity in the semi-specific dataset is quite high and would like to emphasize that the PLS-DA was intended as an exploratory visualization to illustrate global differences in peptide-level profiles, rather than as a definitive biological conclusion.

The PDX analysis and the MSKCC cohort proteomics seem disjointed. Are the nine PDX samples described in section 3.9 derived from the same patients as the eight PDX used for the drug analysis in section 3.10? How are they related to the MSKCC-ICC cohort? I am not convinced of the added value of the PDX proteomics analysis in its current form.

Answer: Indeed, the PDX tissues did not derive from patients in the MSKCC-ICC cohort (or any other cohort described) but have been sourced from commercially available PDX models (CharlesRiver, Freiburg, Germany). We have better highlighted this in the text. The PDX analyzed in the section "Proteomics of ICC PDX Models Enables Insight into Tumor-Stroma Co-Regulation" represent nine different PDX models originating from different donors. Only one of these models, LI-002, was selected for the drug-testing of eft226. Despite the PDX models not being directly related to the cohort patients, we believe that PDX provide a unique opportunity to investigate tumor-stroma interactions via mass spectrometry-based proteomics in-situ and with patient material. In contrast to e.g. tissue co-cultures, tumor-stroma interactions can be investigated within a living organism with the benefit of an easy distinction of human and murine proteins thanks to species-specific differences in protein sequences.

In the conclusion, the authors highlight this as the first large-scale ICC proteomic study with European ancestry-dominated patients. However, the core power of this comparison is not clearly demonstrated. While differences between tumor and NAT protein abundances are shown, this is not surprising.

Answer: We thank the reviewer for this comment. While differences between tumor and NAT protein abundances are indeed expected, the main strength of our study lies in the identification of two robust proteomic subtypes. Our classifier reliably assigns samples to these subtypes across independent cohorts and demonstrates predictive value for survival, highlighting its potential utility beyond descriptive tumor-normal comparisons.

The FU-ICC cohort is used as an external validation dataset, but it is a Chinese-dominated cohort. Do genomic profiles differ between ICC patients of Chinese versus European ancestry? If so, how might this influence the proteome? The statement that "genomic analysis [was] not feasible" is not convincing. More comparative analysis could be performed to evaluate differences in tumor/normal proteomes and survival-associated proteins (section 3.7) between the two cohorts.

Answer: We thank the reviewer for this attentive observation. Indeed, due to the rarity of ICC, comprehensive omics-type studies remain scarce. Still, the impact of patients' geographic location and ancestry on ICC tumors has been studied. One large study compared mutational profiles between 164 ICC tumors of Asian origin to 283 tumors coming from Western patients and discovered significant differences in genes related to DNA repair and immune-checkpoints¹⁹. Different incidence rates and disease etiologies have been reported for ICC across world regions, with East Asia

standing out as the area with the highest ICC rates worldwide²⁰⁻²². This might in large part be linked to differing causes of ICC, which are generally considered to derive rather from obesity, and alcohol and nicotine abuse in Western countries, whereas many cases in East Asia also stem from parasitic infections and e.g. contaminated traditional chinese medicines (in this regard, the Dong et al. study we have integrated as FU-ICC cohort points out high overall mutational burdens in tumors with an aflatoxin and/or aristolochic acid mutational fingerprint)^{13,22}. Hence, at least on the genomic level, ICC are considered to differ between world regions and so far proteomic data is lacking to back this up.

To provide additional context, we also incorporated the UKF-ICC cohort as another Western cohort. In addition to the proteomic analysis, we aimed to perform WES and RNA sequencing. Fusion transcripts unfortunately could not be analyzed due to strongly degraded RNA, resulting in poor RNA sequencing quality. Due to limited sample size eligible for WES, we could not identify a clear mutational pattern in the UKF-ICC cohort.

Still, we refrain from directly comparing proteomic profiles between the tumor tissues in the MSKCC-ICC and FU-ICC cohorts. Although both cohorts can technically be merged using the ComBat algorithm, we cannot assure that underlying data structures are fully comparable. The comparison would invariably be affected by different protocols (e.g. DIA-based in case of MSKCC-ICC vs. TMT-labeling-based in case of FU-ICC), mass spectrometers or separation columns, to name a few. Moreover, data from TANM tissues in the FU-ICC cohort was not made publicly available.

Additionally, the number of proteins identified per patient appears somewhat low compared to the FU-ICC cohort, which raises concerns about the depth and completeness of the proteome coverage in this study.

Answer: We agree and have hence re-measured the entire MSKCC-ICC cohort as described above. We now identify an average of 7,020 proteins per sample, of which more than 5,500 were used for statistical analyses, substantially improving proteome depth and coverage.

Minor Concerns and Suggestions:

Figures are not cited correctly. For example, line 250 cites Figure S2, but it should be S3. Figures are also cited out of order, such as Figure S1C being cited after Figure S2, which is confusing.

Answer: Thank you for pointing this out. We have corrected the figure citations.

The proteomics data analysis requires clearer explanation. It is not clear how the spectral library was generated. The manuscript mentions a pooled sample was used, presumably for DDA, but no details are given about the method or data analysis.

Answer: We remeasured the whole cohort as described above and have now used a predicted spectral library. Details are given in the Supplementary Methods.

Why was a relatively high FDR (10%) used during spectral library generation? Please clarify the method and justification.

Answer: In our initial workflow, we applied a 10% FDR during spectral library generation in order to maximize library size, while maintaining the standard 1% FDR cutoff for peptide-to-spectrum assignments in the actual analyses. We acknowledge, however, that this distinction may cause confusion. To avoid ambiguity and ensure consistency throughout the workflow, we have now recalculated the spectral library using a 1% FDR.

Statistical results are often incomplete. P-values and/or effect sizes/HR are missing in many places, including lines 215 and 222.

Answer: Thank you for this observation, we completed the information.

Statistical analysis approaches are not explained in sufficient detail. For survival analysis (section 3.7), the authors mention using the survival and survminer packages, but what confounding variables were adjusted for? Section 3.7 mentions adjusting for adjuvant radiotherapy, but were adjustments made for ECM/turnover clusters in section 3.5? Why were other variables, such as stage, sex, age, and virus status, not adjusted? These factors were mentioned in the introduction as relevant risk factors. The FDR correction method for proteome-wide survival analysis is also not described.

Answer: We thank the reviewer for this comment. For the Cox-Proportional Hazards survival analysis, we adjusted only for adjuvant radiotherapy, the only independent variable significantly associated with shorter time to recurrence. In section 3.5 (now section "Cox Proportional Hazards Model Highlights Individual Prognostic ICC Proteins in the MSKCC-ICC Dataset", ECM/turnover clusters were the parameters of interest and thus were not adjusted. Other independent variables (sex, age, virus status) were not included as they showed no significant association with outcome. Some variables, such as staging and tumor budding, were significantly associated with survival, but not independent from the proteomic data. To avoid overfitting we thus did not include them in the survival model. Proteome-wide survival analyses were corrected for multiple testing using the Benjamini-Hochberg method.

For the differential abundance analysis of tumor versus normal, were data scaled before fitting the linear model? Did the model include random intercepts for each patient? Were confounding variables considered? How was FDR controlled?

Answer: The input expression matrix was log₂-transformed and median-normalized prior to analysis. No additional scaling was applied before fitting the linear model. The analysis used a fixed-effects design comparing tumor versus normal, without random intercepts for individual patients. Potential confounding variables were not included in the model. False discovery rate (FDR) was controlled using the Benjamini-Hochberg method. We have added this information to the Methods section for clarity.

In the PCA plot (Figure 1F), PC1 explains an unusually high percentage of variance. Were data scaled before running PCA? If not, the difference may be driven by high-abundance proteins.

Answer: Data were log2-transformed and median-normalized prior to PCA, without additional scaling. While PC1 may reflect high-abundance proteins characteristic of tumor or TANM tissue, this variation appropriately captures the separation between tissue types.

Line 246: "PCA of proteome data" is vague. "PCA of protein abundance" may be clearer.

Answer: We thank the reviewer for this comment, we changed the wording as suggested.

Line 248: The interaction between PCA results and gender, age, or stage is not shown in Figure 1F.

Answer: We have now added these variables to the PCA in Figure 1F.

Line 249: The authors state that proteomic clustering was confirmed by MC simulation. It is unclear whether they (1) used hierarchical clustering and found consistency with M3C, or (2) used M3C to determine two clusters, then applied hierarchical clustering. If (1), the consistency between methods should be shown; if (2), the wording should be clarified.

We thank the reviewer for this comment. To clarify, we followed scenario (1): We clustered the cohort using hierarchical clustering and then determined the optimal number of clusters - two in this case - via a Monte-Carlo-Simulation run in hierarchical clustering mode. We have revised the manuscript to clarify this.

Line 264: The authors mention gender. Do they mean gender or sex? If referring to biological characteristics, "sex" is more appropriate.

Answer: We thank the reviewer for pointing this out. We are indeed referring to biological characteristics and thus changed the wording to "sex".

Line 265: Missing p-value cutoff and statistical test used.

Answer: We thank the reviewer for this observation, we have now added p-value and statistical test.

Line 267: Missing p-value.

Answer: We have now added the p-value.

Line 268: Assumed pathway enrichment results, but the description lacks clarity.

Answer: We thank the reviewer for this comment. In this paragraph, we were referring to histological features of the corresponding images as determined by a trained pathologist, not to pathway enrichment. We have revised the manuscript to clarify this point.

Line 272: I believe the authors meant to refer to the "cluster" rather than "cluster association." It is unclear what association is being referenced, and why such an association would be predictive of an outcome. Additionally, it would be helpful to clarify whether confounders were controlled for in this analysis. A significant proportion of patients in the turnover group received adjuvant radiotherapy—were these patients presenting with more aggressive disease, potentially leading both to the decision for adjuvant radiotherapy and to worse survival outcomes?

Answer: We thank the reviewer for this comment. We agree that the term "cluster" is more appropriate than "cluster association" and have revised the text accordingly. The proteomic clusters were defined independently of clinicopathological features, with the classifier distinguishing samples only by proteomic profile (cluster 1 vs. cluster 2), not by outcome. As such, confounders were not incorporated into the clustering procedure, which was based exclusively on protein abundance. We note that a larger proportion of patients in the turnover/proliferation cluster received adjuvant radiotherapy; however, information on the clinical criteria guiding these treatment decisions is beyond accessibility. While this represents a limitation, the observation that proteomic clusters stratify patient outcomes across multiple independent cohorts supports the robustness of the finding.

Line 317: What is the hazard ratio?

Answer: The hazard ratio for the proliferation cluster is 2.1 compared to the ECM cluster.

CPHM - Hazards Ratio for proliferation vs. ECM cluster in the FU-ICC cohort

We have added this information to the manuscript.

Reviewer #4 (Remarks on code availability):

I appreciate the authors' effort to support open science and reproducibility by providing code in the associated GitHub repository. However, the repository appears to only include scripts and R Markdown files for generating the figures, and it seems that some supplementary figures are not covered. Additionally, it appears that all commits were pushed within a two-day window, which means that the authors may have either selectively uploaded certain scripts to the repository specifically for publication or did not employ version control throughout their analysis. I encourage the authors to provide a more comprehensive and transparent workflow, ideally with version-controlled scripts that cover the entire analysis process, which would significantly enhance reproducibility.

Answer: Thank you for this observation and for having a look at our code! We have now included all supplementary figures, too. It is true that the code was specifically prepared for publication. This was for two reasons: first, it was important to us to share reasonably tidy and comprehensible scripts that do not include the manifold detours and trials, which the initial analysis entailed. Secondly, technical reasons prohibited us from using online version control via GitHub. Since our data analysis is performed on a hospital internal server containing patient data, we are prevented from continuous and automated access to online resources such as GitHub

1. Winkler, J., Abisoye-Ogunniyan, A., Metcalf, K. J. & Werb, Z. Concepts of extracellular matrix remodelling in tumour progression and metastasis. *Nat. Commun.* **11**, 5120 (2020).
2. Cosenza-Contreras, M., Huesgen, P. F. & Schilling, O. TermineR: Bioinformatic processing of shotgun proteomics data for the annotation and quantitation of protein termini. *Methods Enzymol.* **719**, 43–66 (2025).
3. Kleifeld, O. *et al.* Isotopic labeling of terminal amines in complex samples identifies protein N-termini and protease cleavage products. *Nat. Biotechnol.* **28**, 281–288 (2010).
4. Dinh, T. J. *et al.* Proteomic analysis of non-muscle invasive and muscle invasive bladder cancer highlights distinct subgroups with metabolic, matrisomal, and immune hallmarks and emphasizes importance of the stromal compartment. *J. Pathol.* **265**, 41–56 (2025).
5. Cosenza-Contreras, M. *et al.* Proteometabolomics of initial and recurrent glioblastoma highlights an increased immune cell signature with altered lipid metabolism. *Neuro-Oncol.* **26**, 488–502 (2023).
6. Clements, O., Eliahoo, J., Kim, J. U., Taylor-Robinson, S. D. & Khan, S. A. Risk factors for intrahepatic and extrahepatic cholangiocarcinoma: A systematic review and meta-analysis. *J. Hepatol.* **72**, 95–103 (2020).
7. Tavolari, S. & Brandi, G. Mutational Landscape of Cholangiocarcinoma According to Different

- Etiologies: A Review. *Cells* **12**, 1216 (2023).
8. Hayashi, A. *et al.* Distinct Clinicopathologic and Genetic Features of 2 Histologic Subtypes of Intrahepatic Cholangiocarcinoma. *Am. J. Surg. Pathol.* **40**, 1021 (2016).
 9. Wang, X.-Y. *et al.* Driver mutations of intrahepatic cholangiocarcinoma shape clinically relevant genomic clusters with distinct molecular features and therapeutic vulnerabilities. *Theranostics* **12**, 260–276 (2022).
 10. Liao, J.-Y. *et al.* Morphological subclassification of intrahepatic cholangiocarcinoma: etiological, clinicopathological, and molecular features. *Mod. Pathol.* **27**, 1163–1173 (2014).
 11. Wang, T. *et al.* Distinct histomorphological features are associated with *IDH1* mutation in intrahepatic cholangiocarcinoma. *Hum. Pathol.* **91**, 19–25 (2019).
 12. Cao, J. *et al.* Intrahepatic Cholangiocarcinoma: Genomic Heterogeneity Between Eastern and Western Patients. *JCO Precis. Oncol.* **4**, PO.18.00414 (2020).
 13. Dong, L. *et al.* Proteogenomic characterization identifies clinically relevant subgroups of intrahepatic cholangiocarcinoma. *Cancer Cell* **40**, 70-87.e15 (2022).
 14. Wang, X. *et al.* Species-Deconvolved Proteomics for In Situ Investigation of Tumor-Stroma Interactions after Treatment of Pancreatic Cancer Patient-Derived Xenografts with Combined Gemcitabine and Paclitaxel. *J. Proteome Res.* **22**, 2436–2449 (2023).
 15. Sueyoshi, K. *et al.* Multi-tumor analysis of cancer-stroma interactomes of patient-derived xenografts unveils the unique homeostatic process in renal cell carcinomas. *iScience* **24**, 103322 (2021).
 16. Hidalgo, M. *et al.* Patient-Derived Xenograft Models: An Emerging Platform for Translational Cancer Research. *Cancer Discov.* **4**, 998–1013 (2014).
 17. Szadvari, I., Krizanova, O. & Babula, P. Athymic nude mice as an experimental model for cancer treatment. *Physiol. Res.* **65**, S441–S453 (2016).
 18. NMRI-nu Immunodeficient Mouse. *Janvier Labs* https://janvier-labs.com/en/fiche_produit/nmri-nu_mouse/.
 19. Cao, J. *et al.* Intrahepatic Cholangiocarcinoma: Genomic Heterogeneity Between Eastern and Western Patients. *JCO Precis. Oncol.* **4**, PO.18.00414 (2020).
 20. Florio, A. A. *et al.* Global trends in intrahepatic and extrahepatic cholangiocarcinoma incidence

from 1993 to 2012. *Cancer* **126**, 2666–2678 (2020).

21. Banales, J. M. *et al.* Cholangiocarcinoma 2020: the next horizon in mechanisms and management. *Nat. Rev. Gastroenterol. Hepatol.* **17**, 557–588 (2020).
22. Woo, S., Kim, Y., Hwang, S. & Chon, H. J. Epidemiology and genomic features of biliary tract cancer and its unique features in Korea. *J. Liver Cancer* **25**, 41–51 (2025).

“Proteomic Characterization of Intrahepatic Cholangiocarcinoma Identifies Risk-Stratifying Subgroups, Proteins Associated with Time-To-Recurrence, and mTOR Effector Molecule EIF4A1 as a Druggable Therapeutic Target”

Response to reviewers - Preface for all reviewers

We thank all reviewers for their comments and suggestions to further improve our manuscript. We addressed all concerns as outlined in the point-by-point response:

Point-by-Point Response

Reviewer 1:

The authors have provided comprehensive responses to the concerns raised by the reviewers, and the authors are to be commended for conducting additional mass spectrometry data acquisition runs using a newer model mass spectrometer to increase the proteomic depth of their MSKCC-ICC cohort. The quality of the manuscript has improved significantly.

Answer: Thank you very much for the encouragement.

On clarifying question and one suggestion:

1. Panel A of the figure on pg. 1 of the response to reviewers refers to “Pearson correlation R2.” It is unclear what statistical parameter the authors are referring to here given that the Pearson correlation is represented by (r), not R2. Similarly, on pg. 2, of their response, the authors mention, “Pearson correlation of R2 = 0.81, whereas “r” is mentioned in the correlation plots below the text.

Answer: Thank you for this comment. We apologize for the confusion, we originally labeled the statistic as “Pearson correlation R².” We have corrected this to “Pearson correlation coefficient R” in all figures and text of our first response to the reviewers (Panel A of the figure on pg.1, text on pg.2).

2. Consider adding the figures on pg. 5 of the response to the concerns raised by the reviewers to the Supplementary Data. The figures provide informative data re: the association between tumor cellularity vs. PFS and OS.

Answer: Thank you for this suggestion, we added the mentioned figure as Supplementary Figure 6C and briefly described it in the main text (see section “ICC Tumor Proteomes Form

Two Clusters with Strong Proteomic Differences and Divergent Times to Recurrence in the MSKCC-ICC Cohort”, ll. 303-305).

Reviewer 2:

I have carefully reviewed the authors' revised manuscript and their point-by-point response. While I appreciate the authors' additional experiments and revisions, several important issues remain insufficiently addressed, as detailed below.

Answer: Thank you for the thoughtful re-evaluation of our manuscript.

1. The authors reanalyzed their peptide samples using the TimsTOF, which increased the number of identified proteins from ~2,000 to over 7,000. This represents a substantial improvement in proteome coverage. However, the subsequent analyses and biological conclusions remain essentially identical to those presented in the previous version. Given this significant increase in data depth, one would expect additional findings or refined biological insights into ICC. Otherwise, this raises the question of whether a much lower proteome coverage would have been sufficient to address the same biological questions. The current revision does not adequately explain why such a dramatic increase in protein identification did not lead to new or strengthened conclusions.

Answer: We appreciate your comment. Although re-measurement on the timsTOF substantially increased proteome coverage (from ~2,000 to >7,000 proteins), the overall biological conclusions remained consistent because the two datasets were generated from the same samples. Their high correlation underscores the robustness and reproducibility of our mass-spectrometry-based approach. Importantly, the expanded dataset did not leave the analysis unchanged; rather, it enabled us to refine and strengthen several key findings:

First, the increased proteomic depth revealed additional differentially abundant proteins between tumor and TANM tissue and across clusters, confirming and extending our previous findings. For example, within the second proteomic cluster, we now observe a more pronounced signature of DNA replication, which was not visible in the first dataset and led us to rename this cluster to better reflect its biological characteristics (see section “ICC Tumor Proteomes Form Two Clusters with Strong Proteomic Differences and Divergent Times to Recurrence in the MSKCC-ICC Cohort”, ll. 260-264 and Figure 3). Similarly, the tumor vs. TANM comparison now shows stronger enrichment of DNA- and RNA-replication signatures (see section “Tumor and TANM Tissues Show Highly Divergent Proteome Profiles”, ll. 230-232 and Figure 2).

The Cox Proportional Hazards model analysis benefited substantially from the deeper coverage, identifying nearly five times more proteins associated with time-to-recurrence than in our first analysis. Accordingly, the subsequent gene set enrichment analysis of these protein

hits revealed that especially proteasome-mediated catabolic-processes were correlated with a bad prognosis. In contrast, proteins related to oxidoreductive processes were linked to a better prognosis (see section “Cox Proportional Hazards Model Highlights Individual Prognostic ICC Proteins in the MSKCC-ICC Dataset”, ll. 332-336 and Figure 5).

Furthermore, our semi-specific analysis was refined by the increased proteome depth, now identifying a higher proportion of semi-specific peptides in the TANM samples compared to tumor tissue (see section “Semi-specific Peptides are Increased in TANM and Differ Between Clusters”, ll. 464-468 and Figure 6).

Although our main conclusions remain nearly unchanged, we view this consistency as reassuring: the new dataset confirms the original biological findings while enabling a more detailed and nuanced interpretation.

2. The authors' point-by-point responses are largely textual and do not provide corresponding figures, figure legends, or manuscript excerpts to demonstrate where and how each issue was addressed. This makes it difficult to evaluate whether and to what extent the reviewers' comments were incorporated into the revised manuscript. For transparency and clarity, the authors should clearly indicate the exact locations of revisions in the text and include updated figures or supplementary materials where relevant.

Answer: We thank the reviewer for this helpful comment. In this revised version of our previous and current point-by-point responses, we have clearly indicated the locations of all revisions in the text and figures referring to the current clean version of our manuscript. We hope that these adjustments make it easier to verify how each comment has been addressed. We would also like to emphasize that all changes to the manuscript can be viewed in the tracked-changes version of the revised document.

3. The authors claim that the role of EIF4A1 in ICC is being reported for the first time. However, recent literature has already described this association — for example: Mi W, Cigliano A, Galleri G, et al. Targeting EIF4A1 is effective against human intrahepatic cholangiocarcinoma. *JHEP Rep.* 2025;7(7):101416.

Furthermore, the study does not provide new mechanistic insights into EIF4A1's role in ICC, which weakens the novelty of this finding.

Answer: We thank the reviewer for bringing this recent study to our attention. We welcome the fact that additional work further supports the relevance of EIF4A1 and its inhibition in intrahepatic cholangiocarcinoma. We have now cited this publication and revised the text accordingly (see section “The mTOR eEffector EIF4A1 Is Strongly Enriched in Tumors of Both Clusters and its Inhibition Reduces Tumor Growth in ICC PDX”, ll. 545-551). While we acknowledge that EIF4A1 has recently been implicated in ICC, our study provides complementary and novel evidence, including the first characterization of EIF4A1 using patient-derived xenograft (PDX) models and the first evaluation of its combination with gemcitabine in ICC cell lines. We believe that these additions strengthen the translational

significance of our findings and further highlight EIF4A1 as a promising therapeutic target in ICC.

4. The authors state that “the PDX models have particular value in corroborating the origin (tumor vs. stromal cells) of the various matrisome components.” However, it remains unclear what specific new conclusions were derived from these experiments. The description is rather general, and neither the main text nor the figures provide sufficient detail to support this claim.

Answer: Thank you for this comment. The PDX analysis provided several insights, which are shown in Figure 7C and described in the corresponding text in the section “Proteomics of ICC PDX Models Enables Insight into Tumor-Stroma Co-Regulation”, ll. 515-532. The key conclusion derived from these data is that the major matrisome and immune-related signatures observed in the ICC cohort likely originate from the invading stroma, whereas the translational and metabolic signatures derive primarily from tumor cells. This implies tumor-stroma interactions are a key determinant for recurrence-free survival. Importantly, several murine ECM components in the PDX models corresponded to proteins defining cluster 1 in the human ICC cohort.

5. Each figure title is currently written as a methodological phrase, rather than summarizing the main finding or conclusion of the figure. This makes it difficult for readers to follow the logical flow of the study or grasp the key results.

Answer: Thank you for this suggestion. We agree that figure legends should guide the reader, but we also believe that detailed conclusions are best placed in the main text, where they can be properly contextualized. Our current legends are written to clearly state the content of each panel, while the interpretation of the results is provided in the corresponding Results section.

Reviewer 3:

Answer: Thank you for your support.

Reviewer 4:

I really appreciate the authors taking the time to perform new experiments and analyses. I am very glad to see that the manuscript has significantly improved, especially in terms of data quality and interpretability.

Answer: Thank you very much for this feedback.

I now like the semi-specific peptide analysis even better. With the new figures, it is now clearer that certain proteins undergo increased proteolytic digestion, particularly in the proliferation-associated cluster. This is a fascinating observation. It raises many interesting follow-up questions that, while outside the scope of this current paper, could inspire future work.

Answer: Thank you for this encouraging observation. We agree and are excited to look deeper into this data as well.

One suggestion I would offer is to explore whether this ECM vs. proliferation tumor subtype classification can also be observed at the transcriptomic level. Since RNA-seq data is available, it would be informative to ask whether ECM- or proliferation-related genes show elevated expression in their respective clusters. Including a figure or two illustrating this could make the picture of the study even more complete.

Answer: Thank you for this helpful suggestion. RNA-seq data were available for the FU-ICC cohort, while the UKF-ICC cohort could not be included due to insufficient RNA quality (RIN values between 1 and 2.6). We analyzed the transcriptomic data of the FU-ICC cohort and indeed observed patterns consistent with our proteomic clusters: ECM-related genes showed higher expression in samples belonging to the proteomic ECM cluster, whereas proliferation-associated genes were elevated in the proteomic proliferation cluster. These results corroborate our proteome-based subtype classification at the transcriptomic level.

Transcriptomic differences between proteomic clusters. Left: Volcano plot of differentially expressed genes between the ECM and proliferation cluster. Log2-fold changes are derived from an univariable linear regression model. Dots colored in orange (proliferation cluster) or blue (ECM cluster) lie above the Bonferroni-adjusted significance threshold. Top hits from the proteomic differential abundance analysis are named, red label indicated diverging direction change in the proteomic data. Right: Overrepresentation analysis of differentially expressed genes between clusters. Top 5 terms per cluster are shown. Significance was determined by a permutation test with false-discovery rate (FDR)-based multiple testing correction.

We added this information to Supplementary Figure 8, panels G and H, and described it briefly in the section “A Classifier for ICC Clusters Detects Similar Proteomic Motifs and Survival Outcome in the FU-ICC Cohort”, ll. 405-407.

The only major issue I would still like the authors to address is the lack of detail in the WXS methods section. Please provide specifics such as the capture kit used, the reference genome version (e.g., GRCh38 p14?), the alignment tool, the somatic mutation caller, filtering criteria, and annotation algorithm. The phrase “standard criteria” is way too vague.

Answer: Thank you for the comment. We used the Agilent v6 capture kit, the reference genome GRCh38 p.13, BWA for alignment and MuTect and Strelka for calling somatic variants. SNPs were filtered for a minimum coverage of 10 reads, variant allele fraction (VAF) > 5%, and ≥ 5 supporting reads in the tumor. Indels were filtered for VAF > 10%. We listed most details regarding the WES workflow (capture kit, alignment tool, somatic mutation caller, filtering criteria, and annotation algorithm) in the Supplementary Methods (see Supplementary Methods “DNA extraction and WES”, ll. 152-159, and “WES Data Analysis”, ll. 278-285). We also extended our main Methods section to include the most relevant parameters as well (ll. 161-165).

Additionally, when discussing mutation burden (e.g., “~30 mutations per tumor”), the authors should report TMB as the number of mutations per Mbp of the capture region. The number of somatic mutations is dependent on the capture design.

Answer: Thank you for this helpful suggestion. The mean TMB was 0.73 mutations per mb of the captured coding regions. We have added this information into the section “Classification of the UKF-ICC validation cohort reveals matching prognostic clusters”, ll. 451-453 and included the information on how we calculated the TMB in the Supplementary Methods, ll. 284-285.

Reviewer #4 (Remarks on code availability):

Seems like only analysis code for figure generation was included.

Answer: Thank you for this comment. We revisited the repository to verify completeness and can confirm that all analysis and figure-generation code is available online.

Intrahepatic cholangiocarcinomas (ICCs) are rare cancers that originate in bile ducts and ductules within the liver. These cancers have non-specific symptoms and there is a lack of reliable, noninvasive prognostic biomarkers. Using a mass spectrometry-based proteomic approach, the authors analyzed 80 treatment-naïve ICC tumors, 77 adjacent non-malignant tissues, and 9 PDXs. Two proteomic sub-clusters, extracellular matrix (ECM) and mRNA/protein turnover, exhibited distinct times-to-recurrence (TTR). These prognostic proteomic clusters were validated using a separate dataset, and EIF4A1 inhibition was identified as a potential strategy to mitigate ICC progression.

The authors generated a significant amount of data for this study; however, there are concerns regarding the data quality and its subsequent impact on the biological interpretation of the data. Several major and minor concerns need to be addressed before this reviewer would consider recommending the manuscript for publication.

Major

1. The font sizes in several of the figures (Fig. S1A & B, 5A & E-F, S3, S5, S7) are too small, which consequently precludes an accurate assessment of the presented data. If the font sizes in the figures cannot be reasonably enlarged, consider reducing the content displayed in each figure to only include the most salient aspects.
2. How do the eFT226 and Gemcitabine concentrations used to treat the ICC cell lines relate to IC₅₀ values?
3. It does not appear that the authors determined the tumor cellularity of the ICC patient tissues. This is a common drawback of the -omic analysis of bulk tissues.
4. A log fold-change cut-off of 0.5 was used for the data presented in section 3.7 regarding the prognostic ICC proteins related to progression-free survival. This is a considerably low cut-off (fold change of ~1.4) to determine whether a protein is “significantly enriched”.
5. The interpretation of semi-tryptic peptides as evidence of “endogenous proteolytic processing events” is dubious. No evidence is provided to enable the differentiation of biologically relevant “endogenous proteolytic processing events” from technical/analytical artefacts. The experimental design for this study was not set up to enable the analysis of endogenous proteolytic processing events.
6. Fig. 2A: There are significant concerns regarding the validity of the number of differentially regulated proteins; 1,368/1,700 = 80%! If the data were normally distributed, only 32%, 4.5%, or 0.3% of the proteins would have a relative abundance exceeding the mean +/- 1 sd, 2 sd, or 3 sd, respectively. It is possible that the protein profiles of TANM vs. tumor are markedly different; however, one would not expect 80% of the proteins to be differentially abundant.
7. It is unclear why the authors identified >2x more human proteins when analyzing the PDX tissue (3,168) vs. the patient tissue (1,368) if the sample processing and analytical methods were indeed similar. The patient tissues and the PDX tissues were FFPE specimens.

Minor:

1. Figure 1E is overly-convoluted/complicated and no key is provided explaining the symbols in the middle of the PCA plots. Is it important to have a unique symbol representing each distinct sample?

2. Fig. S2: Add a legend for the red-blue color gradients/scales.
3. In the legend for Fig. 8, “dotted lines” are referred to as indicating treatment time-points, but they do not appear in the figure. Moreover “grey arrows” are also mentioned as indicating treatment time points.
4. Titles should be included with each Supplementary Table.